# Dietary nitrate supplementation mitigates age-related changes at the neuromuscular junction in mice

Maira Rossi[1] , Lucrezia Zuccarelli[2] , Lorenza Brocca[1] , Cristiana Sazzi[1] , Clarissa Gissi[2] , Paola Rossi[3] , Bruno Grassi[2] , Simone Porcelli[1,4] , Roberto Bottinelli[1,5] and Maria Antonietta Pellegrino[1,6]

[1] *Department of Molecular Medicine, University of Pavia, Pavia, Italy*
[2] *Department of Medicine, University of Udine, Udine, Italy*
[3] *Department of Biology and Biotechnology, University of Pavia, Pavia, Italy*
[4] *IRCCS Fondazione Policlinico San Matteo, Pavia, Italy*
[5] *IRCCS Mondino Foundation, Pavia, Italy*
[6] *Interdipartimental Centre for Biology and Sport Medicine, University of Pavia, Pavia, Italy*

Handling Editors: Richard Carson & Christoph Centner

The peer review history is available in the Supporting Information section of this article (https://doi.org/10.1113/JP287592#support-information-section).

**Abstract figure legend** Ageing is associated with a decline in muscle mass and strength, namely sarcopenia, significantly impacting the quality of life in older individuals. Different strategies are being explored to counteract these detrimental effects. In the present study, we investigated a non-invasive nutritional approach: the impact of 2 months of nitrate supplementation in 24-month-old male mice. Compared to young mice, old mice exhibited reduced muscle fibres cross-sectional area (CSA), higher neuromuscular junction (NMJ) fragmentation, reduced innervation and higher oxidative stress, all indicative of age-related muscle deterioration. Nitrate supplementation in aged mice led to muscle mass recovery, improved NMJ morphology, enhanced innervation and reduced oxidative stress. Our findings suggest that nitrate supplementation may have a beneficial effect on ageing muscles, providing a potential nutritional approach to mitigate age-related muscle decline. Created with BioRender.com

**Abstract** In ageing, denervation and neuromuscular junction (NMJ) instability occur alongside mitochondrial alterations and redox unbalance, potentially playing a significant role in the process. Moreover, the synthetic pathway was shown to be critical for proper innervation and NMJ stability. Nitric oxide (NO) modulates redox status, mitochondrial function and the synthetic pathway. Its bioavailability declines with age. We hypothesize that nitrate supplementation could counteract age-related neuromuscular alterations. We compared young (Y) (7 months old), old (O) (24 months old) and old mice supplemented daily with 1.5 mM inorganic $NaNO_3$ dissolved in drinking water for 8 weeks (ON) (24 months old). Compared to Y, O mice displayed impaired NO signalling and transport (lower phosphorylated-neuronal NO synthase and sialin content); greater nitrosative and oxidative stress (higher 3-nitrotyrosine levels and protein carbonylation); lower glutathione peroxidase (GPX antioxidant enzyme); smaller muscle fibres; and larger muscle fibrosis. NMJ integrity was impaired, exhibiting age-related alterations such as larger fragmentation, lower overlap, larger endplate areas and lower compactness. Consistently, greater expression of denervation-associated markers (Gadd45$\alpha$, MyoG, RUNX1, AChR$\gamma$ and NCAM1) and higher NCAM1+ fibres percentage suggested denervation. Importantly, mitochondrial content, dynamics and function were unchanged. Compared to O, ON mice showed improved NO bioavailability in muscle (higher nitrate-nitrite concentration); lower fibrosis and improved muscle fibre size; higher phosphorylation of P70S6K and S6, downstream factors of Akt/mammalian target of rapamycin synthetic pathway; lower oxidative stress (lower carbonylated proteins and mitochondrial hydrogen peroxide production, higher GPX protein levels); reverted age-related alterations of NMJ morphology; and lower percentage of NCAM1+ fibres. Nitrate supplementation could be a therapeutic strategy to counteract muscle decline with ageing.

(Received 16 September 2024; accepted after revision 6 February 2025; first published online 25 February 2025)

**Corresponding author** M. Rossi: Department of Molecular Medicine, University of Pavia, Via Forlanini 6, 27100 Pavia, Italy. Email: maira.rossi@unipv.it

## Key points

- Ageing leads to instability at the neuromuscular junction (NMJ), which is crucial for muscle size and function, ultimately giving rise to denervation and muscle fibres loss.
- Mitochondrial function, redox status and activation of synthetic pathway are critical processes for proper muscle innervation and stability of the NMJ.
- Nitric oxide was shown to modulate intracellular processes involved in NMJ stability such as balance of reactive oxygen species, mitochondrial function and protein synthesis. Its bioavailability decreases with ageing.
- Our study shows that nitrate supplementation in old mice improved redox balance, enhanced the anabolic pathway and stabilized nerve–muscle interactions, suggesting a potential strategy to mitigate the neuromuscular decline associated with ageing.

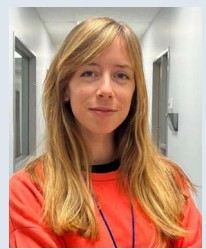

**Maira Rossi** earned her PhD in Translational Medicine at University of Pavia (Italy) in 2022. During her PhD and subsequent postdoctoral fellowships in the laboratory coordinated by Maria Antonietta Pellegrino and Roberto Bottinelli, she investigated neuromuscular and molecular mechanisms underlying muscle wasting in ageing, loading, unloading and disease through morphological and molecular approaches. Subsequent to July 2023, she has been a visiting researcher at Sanford Burnham Prebys Medical Discovery Institute (La Jolla, CA, USA) in Pier Lorenzo Puri's laboratory to further her approach to muscle plasticity through single cell and single nucleus transcriptomics and spatial -omics technologies. In the future, she aims to address the challenges of treating muscle-wasting conditions to improve clinical outcomes of affected individuals.

## Introduction

Ageing leads to a progressive loss of muscle mass and strength, a condition that has significant implications for human health and can be clinically diagnosed as sarcopenia (Cruz-Jentoft et al., 2019). Although the mechanisms underlying sarcopenia have been extensively studied, they remain incompletely understood (Larsson et al., 2019).

Denervation is considered a major driver of muscle deterioration in ageing. Reinnervation by nearby axon terminals can initially compensate muscle fibres denervation, maintaining muscle mass for years. However, in advanced age, reinnervation eventually fails, giving rise to permanent denervation and fibre atrophy and death, causing loss of muscle mass and strength (Hepple & Rice, 2016). The neuromuscular junction (NMJ), the crucial site of nerve–muscle interaction, plays a key role in such processes. Indeed, in ageing, NMJ is characterized by a loss of the typical pretzel-like structure and a structural fragmentation, as observed in both rodents (Chai et al., 2011; Cheng et al., 2013; Deschenes et al., 2010; Gutmann & Hanzlikova, 1966) and humans (Oda, 1984; Wokke et al., 1990). The consequences of NMJ fragmentation on signal transmission are still debated (Willadt et al., 2016). Nevertheless, a correlation between denervation and muscle atrophy was established in both physiological (Aare et al., 2016) and transgenic (Aare et al., 2016; Deepa et al., 2019) mouse models of ageing.

Mitochondrial alterations and reactive oxygen species (ROS) production are considered potential candidates of NMJ destabilization and denervation (Anagnostou & Hepple, 2020). Although the classical theory of ageing, which posits that ROS from the respiratory chain lead to mitochondrial DNA mutations, is widely questioned (Larsson et al., 2019; Wang & Hekimi, 2015), mitochondria and ROS are still considered to be major players in ageing (Anagnostou & Hepple, 2020; Larsson et al., 2019; Rygiel et al., 2016). Mitochondria, abundant in both sides of the NMJ (Rygiel et al., 2016), generate ROS, especially when alterations in their dynamics occur (Picard et al., 2010; Romanello & Sandri, 2010). NMJ instability in ageing may relate to ROS accumulation. For example, transgenic $SOD1^{G93A}$ mice, which harbour a non-functional mutation in the SOD1 gene, encoding for the antioxidant enzyme superoxide dismutase 1 (SOD1), show oxidative stress, progressive muscle atrophy, decreased muscle strength, mitochondrial dysfunction and increased NMJ fragmentation (Dobrowolny et al., 2008, 2018). Notably, overexpression of catalase, an antioxidant enzyme that neutralizes hydrogen peroxide ($H_2O_2$) in mitochondria, was shown to prevent NMJ disruption and muscle wasting in SOD1KO mice (Xu et al., 2021).

Ageing is characterized by nitric oxide (NO) decline (Di Massimo et al., 2006; Donato et al., 2018) as a result of impaired nitric oxide synthase (NOS) activity or defects in NO metabolism and storage. Increasing NO bioavailability may attenuate the ageing process. NO is a small molecule involved in a plethora of signalling pathways. It is synthesized by endogenous enzymes (NO synthases, NOS) that convert L-arginine into the bioactive end-product NO. More recently, it was demonstrated that dietary nitrate ($NO_3^-$) consumption can increase NO bioavailability through a NOS independent pathway. When ingested, $NO_3^-$ is rapidly absorbed in the stomach and enters the enterosalivary circulation. Here, oral commensal bacteria can further reduce $NO_3^-$ to nitrite ($NO_2^-$) (Duncan et al., 1995), which can be subsequently metabolized to NO in the circulation or peripheral tissues (Lundberg & Govoni, 2004). Several studies have demonstrated that $NO_3^-$ supplementation modulates NO homeostasis (Piknova et al., 2022) and improves skeletal muscle perfusion (Bailey et al., 2015; Ferguson et al., 2013a, b), force production (Hernandez et al., 2012; Stamler & Meissner, 2001) and mitochondrial bioenergetics in both mice (Brunetta et al., 2022) and humans (Larsen et al., 2011). Additionally, $NO_3^-$ supplementation enhances NO bioavailability (Upanan et al., 2024) and storage in muscles (Kiani et al., 2022; Nyakayiru et al., 2017). More importantly, $NO_3^-$ supplementation was shown to attenuate mitochondrial ROS production and $H_2O_2$ accumulation, ameliorating ADP availability for mitochondrial oxidative capacity in rodent models of impaired muscle metabolism (Brunetta et al., 2022; Petrick et al., 2022). Finally, consistent with the role of NO in modulating mammalian target of rapamycin (mTOR) downstream factors that promote hypertrophy (Ham et al., 2014; Ito et al., 2013), L-arginine was shown to enhance protein synthesis through the Akt/mTOR pathway in C2C12 cell cultures (Wang et al., 2018) and in chicken muscles *in vivo* (Wang et al., 2022). Additionally, dietary $NO_3^-$ enhanced mitochondrial protein synthesis rates in rodents (Petrick et al., 2025). These latter observations are particularly relevant because the mTOR pathway was shown to be critical for NMJ integrity (Baraldo et al., 2020; McCann et al., 2007).

The present study aims to investigate the ability of $NO_3^-$ supplementation to counteract age-related neuromuscular alterations. We hypothesize that $NO_3^-$ would improve NMJ stability by modulating one or more of the following potential determinants: redox balance, mitochondrial features and muscle protein synthesis. To achieve our goal, we comprehensively analysed key sarcopenia parameters, including NMJ integrity, mitochondria (content, dynamics and function), redox balance and related intracellular signalling pathways.

# Methods

## Ethical approval

Animal handling and experimentation were performed in line with approved Institutional Animal Care and Use Committee protocols at the University of Pavia (n. 774/2016PR) and followed the national authority guidelines for the detention, use and ethical treatment of laboratory animals. The study conformed to the ethical principles and regulations outlined by *The Journal of Physiology* (Grundy, 2015).

## Animals

Wild-type C57BL/6 male mice (Charles River Laboratories, Wilmington, MA, USA) were generously provided by Professor Paola Rossi. Mice were housed in a conventional laboratory animal facility at constant temperature and controlled humidity, under a 12:12 h light/dark photocycle to ensure natural circadian rhythm. Three groups were assessed: young mice (Y) (7 months old), old mice (O) (24 months old) and old mice supplemented with $NaNO_3$ for 2 months (ON) (24 months old). Old mice were randomly assigned to the latter two groups. All mice had full access to food and water *ad libitum*. Seven to eight animals per group were used.

## $NO_3^-$ supplementation

$NO_3^-$ supplementation consisted of daily administration of 1.5 mM $NaNO_3$ (∼8 µmol day$^{-1}$ nitrate; 1065370500; Sigma-Aldrich, St Louis, MO, USA) dissolved in drinking water, and calculated by the average daily consumption of 22-month-old mice (∼6 mL day$^{-1}$). The dosage was initially selected based on the study of Hernandez et al. (2012). It was adjusted to 1.5 mM, considering the mice body weight and recent evidence suggesting age-related modulation of oral nitrate-reductase activity, which can lower nitrate/nitrite blood levels (Ahmed et al., 2021). Mice were supplemented for 2 months (months 22–24). $NO_3^-$ solutions were prepared daily by dissolving a concentrated stock solution of $NaNO_3$ in drinking water. Aliquots of the concentrated stock were stored at −20°C to prevent $NO_3^-$ degradation (Corleto et al., 2018) and thawed daily to prepare the drinking solution. Throughout the experimental period, weight and water consumption were monitored weekly.

It is expected that, at the same dosage, similar plasma nitrate levels would be achieved in both humans and mice, whereas nitrite levels would be higher in humans. This expectation is based on the findings of Montenegro et al. (2016), who compared plasma nitrate and nitrite levels in humans and mice following the administration of the same nitrate dose.

## Sample collection

Animals were fasted 2 h prior to death. Mice were weighted and subsequently killed by cervical dislocation, and the muscles were immediately dissected and processed for different analyses as described below. Given the extensive number of determinations carried out to obtain a comprehensive picture of the phenomena and the relatively small size of mice muscles, it was not feasible to perform all analyses on the same muscle. Therefore, to limit the number of animals used in accordance with the 3Rs principle, muscles were optimized for the different analyses as follows: extensor digitorum longus (EDL) was selected for the study of NMJ morphology under confocal microscopy because of its small size, which is suitable for this analysis; concentration of NO-related compounds was determined in soleus; tibialis anterior (TA) and gastrocnemius (Gas), which share similarity in terms of muscle phenotype, were used for *ex vivo* mitochondrial function via high-resolution respirometry and *in vitro* morphometric and molecular determinations, respectively. In adult mice, EDL, Gas and TA are fast muscles, whereas soleus is a mainly slow muscle (Desaphy et al., 2010). Specifically, EDL and TA express only myosin heavy chain (MHC)-2B and 2X isoforms, with MHC-2B relative content exceeding 90% in EDL and 80% in TA. Gas is mainly composed of MHC-2B isoform (∼90% relative content) with a low content in MHC-2X and 2A (less than 10% combined relative content) and a very low content in MHC-1 (1–3% relative content). MHC-1 content is clustered in the deeper portion of the muscle. By contrast, soleus is mainly composed of MHC-1 (∼40% relative content) and MHC-2A (∼50% relative content), with minor amounts of MHC-2X. In terms of energy production, it is widely assumed that a coordinated expression of MHC isoforms and metabolic enzymes occurs in skeletal muscle fibres, with slow fibres, containing MHC-1, exhibiting higher oxidative metabolism than fast fibres (Reichmann & Pette, 1982). Indeed, an inverse relationship between MHC isoform content and metabolic enzymes was observed in rat, hamster and human, with slow muscle fibres containing MHC-1 showing higher oxidative enzymes content compared to fast fibres containing MHC-2A, 2X and 2B (Delp & Duan, 1996; Mattson et al., 2002; Murgia et al., 2021; Rosser & Hochachka, 1993). Interestingly, in rat and hamster, the highest citrate synthase (CS) activity was found in MHC-2A fibres, with activity decreasing in the order MHC-2A > MHC-1 > MHC-2X > MHC-2B. Moreover, no large differences in CS activity were found between fast muscles (e.g. EDL and TA) and soleus, the

slowest muscle in rodents (Luedeke et al., 2004; Pellegrino et al., 2004). We can assume that the CS activity in the fast muscles used in the present study, mainly composed of MHC-2X and MHC-2B containing fibres, was similar.

## $NO_3^-$ concentration

To quantify NOx concentration in the muscles, frozen (liquid nitrogen) soleus muscles were homogenized in phosphate-buffered saline (PBS) containing 136 mm NaCl, 2 mm KCl, 6 mm $Na_2HPO_4$, and 1 mm $KH_2PO_4$, pH 7.4) on ice and rapidly centrifuged at 10,000 *g* for 30 min, and supernatant used for quantification. Concentration of NOx was measured using a commercial colorimetric assay kit (780001; Griess Reagent Kit; Cayman Chemical, Ann Arbor, MI, USA), which provides an accurate and convenient method to determine $NO_3^-$ and $NO_2^-$ concentrations. Next, 40 µg of each sample was plated in a microplate reader spectrophotometer (CLARIOstar® Plus; BMG Labtech, Ortenberg, Germany), in duplicate. Both $NO_3^-$ and $NO_2^-$ were quantified, in accordance with the manufacturer's datasheet.

## *Ex vivo* mitochondrial high-resolution respirometry

For the measurements of mitochondrial respiration, the Oxygraph-2k (O2k; OROBOROS Instruments, Innsbruck, Austria) was used and combined with Fluorescence-Sensor Green of the O2k-Fluo LED2-Module for $H_2O_2$ measurement.

The freshly excised TA was washed in ice-cold BIOPS (biopsy preservation solution: 2.77 mm $CaK_2$-EGTA, 7.23 mm $K_2$EGTA, 5.77 mm $Na_2$ATP, 6.56 mm $MgCl_2 \cdot 6H_2O$, 20 mm taurine, 15 mm $Na_2$ phosphocreatine, 20 mm imidazole, 0.5 mm dithiothreitol and 50 mm MES, pH 7.1 at 0°C) and properly dissected to obtain two contiguous bundles of well-oriented fibres (∼6 µg). Bundles contiguous to those subjected to determination of mitochondrial respiration were weighed and used for CS activity determinations (see below). The continuity of the bundles used for the two determinations ensured the results could be combined. Muscle fibres were mechanically separated with small fine-tipped tweezers under the microscope, in ice-cold BIOPS. Plasma membrane was permeabilized by gentle agitation for 30 min at 4°C in 2 mL of BIOPS containing 50 µg mL$^{-1}$ saponin. Permeabilized fibres were thereafter washed for 10 min with miR05 [mitochondrial respiration medium: 110 mm sucrose, 60 mm K-lactobionate, 0.5 mm EGTA, 3 mm $MgCl_2$, 20 mm taurine, 10 mm $KH_2PO_4$ and 20 mm HEPES, pH 7.1 at 30°C, and 0.1% bovine serum albumin (BSA) essentially fatty acid free] (Gnaiger et al., 2000), weighed in a balance-controlled scale (Mettler Toledo,

Columbus, OH, USA) and immediately transferred into Oroboros-O2k chambers.

All experiments were performed in 2 mL of miR05 containing myosin II-ATPase inhibitor (blebbistatin, 25 µm, dissolved in 5 mm dimethyl sulfoxide) (Perry et al., 2011) to prevent spontaneous contraction in the respiration medium. Samples were kept at 37°C under hyperoxic conditions. The chamber $O_2$ concentration was maintained between 250 and 450 nmol mL$^{-1}$ (average $O_2$ partial pressure 250 mmHg) to avoid $O_2$ limitation of respiration. The supraphysiological $PO_2$ range maintained throughout the experiment (280–400 µm) is necessary to ensure stable and reliable oxygen flux measurements, preventing oxygen limitation that could cause hypoxic core development in muscle fibres preparations, potentially compromising respiration accuracy (Scandurra & Gnaiger, 2010). Reoxygenation steps were performed during the experiments by injection of pure gaseous $O_2$ when needed. Standardized instrumental and chemical calibrations were performed to correct for back-diffusion of $O_2$ from the various components into the chamber (Pesta & Gnaiger, 2012). Because the Oroboros-O2k set up has two chambers, two samples were evaluated in parallel. Each data point is the resultant of the mean values of duplicates of specimens from the same TA.

Respiration of permeabilized muscle fibres was determined using substrate-uncoupler-inhibitor titration (SUIT) protocols described previously (Makrecka-Kuka et al., 2015; Zuccarelli et al., 2021) with modifications. Glutamate (G) and malate (M) (10 mm and 4 mm, respectively) were added to measure non-phosphorylating resting mitochondrial respiration in absence of adenylates ('LEAK' respiration of Complex I (CI)). Succinate (Succ, 10 mm) was then used to determine the LEAK of both CI and CII. ADP was added at increasing submaximal concentrations until reaching a final concentration of 4 m (saturating for oxygen flux) to obtain maximal ADP-stimulated mitochondrial respiration (OXPHOS capacity) through CI and CII. Stepwise additions of the uncoupler protonophore carbonylcyanide-*p* trifluoromethoxyphenylhydrazone (two steps of 1 µm each) were performed to determine the electron transfer system (ETS) capacity through CI and CII. Rotenone (Rot, 0.5 µm added to inhibit CI) and antimycin A (AmA, 2.5 µm to inhibit CIII and thus mitochondrial respiratory chain) were added for the determination of maximal respiratory uncoupled efficiency and residual oxygen consumption (ROX), respectively. Mitochondrial respiration indices were then corrected for $O_2$ flux resulting from ROX subtraction. Prior to AmA, cytochrome *c* (CytC, 10 µm) was added to evaluate the integrity of the outer mitochondrial membrane, excluding samples with the oxygen flux exceeding 15%.

$H_2O_2$ flux was measured simultaneously with respirometry in the O2k-Fluorometer using the

$H_2O_2$-sensitive probe Amplex® UltraRed (A36006; Thermo Fisher Scientific, Waltham, MA, USA). Then, 10 μM Amplex® UltraRed (AmR), 1 U mL$^{-1}$ horseradish peroxidase (HRP) and 5 U mL$^{-1}$ superoxide dismutase (SOD) were added to the chamber prior to the SUIT protocol. Calibrations were performed with $H_2O_2$ repeatedly added at 0.1 μM steps, in accordance with the manufacturer's indications (Makrecka-Kuka et al., 2015).

All the chemicals used were purchased from Sigma-Aldrich, in accordance with the protocol suggested by the manufacturer of the Oxygraph-2k (OROBOROS Instruments), unless mentioned otherwise. At the conclusion of each experiment, muscle samples were removed from the chambers, washed in PBS and centrifuged for 10 min at 14,000 $g$ at 4°C, and immediately frozen in liquid nitrogen to be stored at -80°C.

Volume-specific $O_2$ and $H_2O_2$ fluxes were calculated real-time using DatLab software (OROBOROS Instruments) over time. Only the stable portions of the apparent fluxes were selected and artefacts induced by additions of chemicals or re-oxygenations were excluded. All the mitochondrial respiration indices were corrected for $O_2$ flux resulting from ROX. The obtained mitochondrial respiration values had to be normalized by CS activity (see below), taken as an index of mitochondrial mass (Larsen et al., 2012). The results are expressed in pmol s$^{-1}$ mg$^{-1}$ wet weight. Each reported value is the average of duplicate analyses.

## CS activity

CS activity was determined as described by Zuccarelli et al. (2021). Briefly, TA samples were thawed and homogenized in a glass potter (WHEATON®, DWK Life Sciences, Millville, NJ, USA), resuspended 1:50 w/v in a homogenization buffer containing 250 mM sucrose, 20 mM Tris, 40 mM KCl and 2 mM EGTA, with 1:50 v/v protease (P8340; Sigma-Aldrich) inhibitors. Prior to the last of the 20 strokes (at 500 rpm), Triton X-100 (0.1% v/v) was added to the solution. After 30 min incubation in ice, the homogenate was centrifuged at 14,000 $g$ for 10 min. The supernatant was used to evaluate protein concentration and the quantified extracts (5–10–15 μg) were added to each well of a 96-well-microplate. Next, 100 μL of 200 mM Tris, 20 μL of 1 mM 5,5′-dithiobis-2-nitrobenzoate, freshly prepared, and 6 μL of 10 mM acetyl-coenzyme A (acetyl-CoA) were added along with MilliQ water (Merk Millipore, Burlington, MA, USA) to reach a final volume of 190 μL. A background ΔAbs, to detect any endogenous activity by acetylase enzymes, was recorded for 90 s with 10 s intervals at 412 nm at 25°C using an EnSpire 2300 Multilabel Reader (PerkinElmer, Waltham, MA, USA). The ΔAbs was subtracted from the one given after the

addition of 10 μL of 10 mM oxalacetic acid that started the reaction. All assays were performed at 25°C in triplicate. Activity was expressed as nmol min$^{-1}$ (mU) mg$^{-1}$ protein.

## Immunohistochemistry

Gas muscles were frozen in liquid nitrogen. Briefly, serial 10 μm thick transverse sections were obtained from the mid-belly region of Gas using a cryostat (set at –20°C; Leica Systems, Wetzlar, Germany). The choice of such region, where pennation angle is minimal, minimizes the presence of non-orthogonal muscle fibres sections that would artefactually increase cross-sectional area (CSA) values. Additionally, to verify the accuracy of the transverse orientation sections, the form factor ($4\pi A/P^2$) was calculated for individual fibres using ImageJ (NIH, Bethesda, MD, USA). Only fibres with a value greater than 0.7 were included in the CSA calculation (Charifi et al., 2004) (see below). Glass slides were conserved at –20°C for further histological investigations (see below).

**MHC isoforms staining.** Frozen sections were immunostained with the antibody specific for mouse anti-slow MHC isoform (dilution 1:2000; Abcam, Cambridge, UK; catalog. no. ab11083; RRID:AB_297734) or for mouse anti-fast MHC isoforms (dilution 1:1000; Abcam; catalog. no. ab51263; RRID:AB_2297993) in 1% BSA (w/v) in PBS and distributed on the surface of the slides to completely cover each section. After 1 h of incubation at 37°C, slides were washed with PBS. Then, the slides were incubated for 30 min at 37°C with an HRP-conjugated rabbit anti-mouse immunoglobulin G (IgG) antibody (Agilent, Santa Clara, CA, USA; catalog. no. P0260; RRID:AB_2636929), diluted 1:500 in 1% BSA in PBS. The reaction of HRP to the addition of diaminobenzidine (3,3′-DAB, D4293; Sigma-Aldrich) produced a colorimetric reaction. A dehydration step in alcoholic solutions of increasing concentration was performed for a few seconds for each concentration (40% ethanol, 60% ethanol, 80% ethanol, 95% ethanol, 50% ethanol and 50% xylene, and then a long wash in xylene). Finally, coverslips were mounted using the Eukitt® Mountant (09-00100; Bio-Optica, Milan, Italy). Muscle sections were then captured at 10× magnification, using a computerized image analyser, consisting of a camera (Digital Vision, Leica Systems, Wetzlar, Germany) placed on a light microscope (Leica DM/LS). Images were then reconstructed with ImageJ (NIH). The results are reported as a percentage of positive fibres.

**Picrosirius Red staining for fibrotic tissue.** Muscle sections stored at −20°C were dried at room temperature for 10 min before rehydration in alcoholic solutions of decreasing concentrations (xylene 100%, xylene/ethanol

50%, ethanol 95%, 80%, 70% and 50%) and subsequent fixation in a neutral buffered formaldehyde solution 3.7% for 1 h at room temperature. Following incubation in Sirius Red (Direct Red 80, 365548; Sigma-Aldrich) 0.1% (w/v) in a saturated aqueous solution of picric acid (1.3% saturated in water, P6744; Sigma-Aldrich) for 1 h at room temperature, two washes in acidified water (0.5% acetic acid in water) were performed to remove exceeding stain. Dehydration in two changes of 100% ethanol is then required, with a subsequent wash in ethanol/xylene in a 1:1 ratio and final clearing in xylene. Finally, coverslips were mounted on the slides with the Eukitt® Mountant (09-00100; Bio-Optica). Images were captured and acquired as described before and analysed with ImageJ (NIH) as a percentage of the fibrotic area on the total area of the section.

**CSA analysis, centronucleated fibres and neural cell adhesion molecule 1 (NCAM1)+ fibres.** Frozen muscle sections were fixed in cold methanol (–20°C) for 10 min and further permeabilized with a solution based on PBS and 2% Triton X-100 (v/v) for 30 min. Then, sections were incubated for 1 h in blocking solution (2% BSA w/v and 2% normal goat serum v/v in PBS) in a humidified chamber. After, they underwent incubation with primary antibodies against mouse anti-NCAM1 (dilution 1:500; Abcam; catalog. no. ab9018; RRID:AB_306945) and rabbit anti-dystrophin (dilution 1:500; Abcam; catalog. no. ab15277; RRID:AB_301813) in blocking solution, overnight at 4°C in humid chamber. Then, conjugated anti-mouse Alexa Fluor$^{TM}$ 594 (Cell Signaling Technology, Danvers, MA, USA; catalog. no. 8890; RRID:AB_2714182) and anti-rabbit Alexa Fluor$^{TM}$ 488 (Cell Signaling Technology; catalog. no. 4412; RRID:AB_1904025) were diluted 1:500 in blocking solution and incubated for 1 h at room temperature, in humid chamber and under dark conditions. Repeated washes in PBS were performed and nuclei were counterstained for 5 min with 4′,6-diamidino-2-phenylindole (DAPI) (Cell Signaling Technology; catalog. no. #4083) diluted 1:5000 in PBS. Negative controls (without the primary antibody incubation) were performed in parallel with the standard protocol to demonstrate the specificity of the fluorescence signals. Slides were mounted with a glycerol-based anti-fade mounting medium and images were acquired at $10\times$ magnification with a fluorescence microscope (Olympus Microscopes, Heidelberg, Germany), elaborated and analysed with the support of ImageJ (NIH). Total numbers of fibres, centralized nuclei and fibres positive for NCAM1 signal were counted and analysed for comparison between Y *vs* O and O *vs* ON. Moreover, the CSA of fibres was measured using ImageJ (NIH) through the analysis of dystrophin borders and expressed as μm$^2$.

## Confocal microscopy for NMJ morphology

EDL muscle was used for the evaluation of the NMJ structure in a preserved whole-mounted tissue. Snap-frozen muscles (in liquid nitrogen and stored at −80°C) were divided in half along the length to reduce the thickness of the tissue and allow a deep investigation. Prior to defrost, the half-muscles were pinned in silicon-based dishes and plunged within cold 4% paraformaldehyde in PBS for 1 h at room temperature to allow fixation of all the structures. Tissue was permeabilized with 4% Triton X-100 (v/v) for 90 min and subsequently incubated in a blocking solution of 4% BSA (w/v) and 2% Triton X-100 (v/v) in PBS, at room temperature for 2 h. Then, primary antibodies for the visualization of presynaptic structures were diluted in blocking solution and incubated for 48 h at 4°C. In detail, rabbit anti-neurofilament L (Cell Signaling Technology; catalog. no. 2837; RRID:AB_823575) and mouse anti-synaptophysin (Cell Signaling Technology; catalog. no. 9020; RRID:AB_2631095) were diluted 1:400 and 1:200, respectively. After, samples were incubated in blocking solution for 30 min prior to be incubated with secondary antibodies. Anti-mouse and anti-rabbit Alexa Fluor$^{TM}$ 488 (Cell Signaling Technology; catalog. no. 4408; RRID:AB_10694704; Cell Signaling Technology; catalog. no. 4412; RRID:AB_1904025) were both diluted 1:500, together with TRITC-conjugated $\alpha$-bungarotoxin ($\alpha$-BTX; dilution 1:1000; Thermo Fisher Scientific; catalog. no. T1175) in blocking solution, and used for 120 min of incubation at room temperature under dark conditions. At the end, samples were plunged in DAPI (Cell Signaling Technology; catalog. no. #4083) diluted 1:5000 in PBS for 10 min to mark the nuclei, and then repeatedly washed in PBS. Finally, muscles preparations were stored for the short term in PBS at 4°C protected from excessive light exposure prior to imaging.

Samples were then mounted on glass slides in VECTASHIELD Vibrance® Antifade Mounting Medium (H-1700; Vector Laboratories, Newark, CA, USA) and acquired on a TCS SP8 confocal laser scanning microscope (Leica Microsystems). Confocal settings were optimized to achieve the best compromise between image quality and acquisition rate: 8 bit depth, $512 \times 512$ frame size, $\times 20$ magnification, $\times 2$ zoom and 1 μm z-stack interval, with sequential image acquisition to minimize bleed through (red channel: 543 nm excitation, 565–615 nm collection; green channel: 488 nm excitation, 500–550 nm collection). NMJs that were partially oblique to the field of view were only included if the oblique portion constituted less than ∼10% of the total area. All image analyses were performed on maximum intensity projections of the z-stacks, using ImageJ (NIH) and related plugins (BinaryConnectivity; Landini, 2008). Specific analyses of NMJ morphology and characteristics were assessed as previously finely described by Jones et al. (2016). At the

end, three mice were analysed per experimental group. An average of 29 NMJ structures were analysed per mice (i.e. a total of 70–110 single NMJ per experimental group).

## Western blot analysis

Frozen muscles (in liquid nitrogen and stored at −80°C) were pulverized and immediately homogenized in lysis buffer (20 mM Tris-HCl, 1% Triton X-100, 10% glycerol, 150 mM NaCl, 5 mM EDTA, 100 mM NaF, and 2 mM $Na_4P_2O_7$) supplemented with protease inhibitor 5X (Protease Inhibitor Cocktail; P8340), phosphatase inhibitors 1X (P0044) and 1 mM phenylmethanesulfonyl fluoride (78830). All reagents were purchased from Sigma-Aldrich. The homogenate was incubated on ice for 40 min and then centrifuged at 18,000 *g* for 20 min at 4°C. Protein concentration was determined using the RC DC™ protein assay kit (#5000122; Bio-Rad Laboratories, Inc. Hercules, CA, USA). Equal amounts of muscle samples were loaded on gradient Precast gels purchased from Bio-Rad (Any kD; #4569034 or 4–20% gradient, #4561094; Bio-Rad Laboratories, Inc.). Then, 40 μg of lysates was standardly used for phosphoprotein evaluations, 15 μg was prepared for mono- and polyubiquitinated proteins, and 5 μg was used for mitochondrial complex evaluation (OXPHOS). Denaturation was performed at 95°C for 5 min prior to the electrophoretic run. As a result of the elevated number of samples under investigation, a reference sample was prepared and loaded in every gel during the same experimental session to make comparison among the samples possible.

After the gel run, proteins were electrotransferred to nitrocellulose or polyvinylidene difluoride (PVDF) membranes at 35 mA overnight. Membranes were stained with Ponceau S (P3504; Sigma-Aldrich) for the evaluation of protein transfer and for normalization analyses. Then, membranes were incubated with a blocking solution consisting of fat-free milk or BSA (for the concentration of milk and BSA, specific for each antibody, see Table 1) dissolved in Tris-buffered saline with Tween-20 (TTBS) 1X (0.02 M Tris, 0.05 M NaCl and 0.1% Tween-20) for 2 h at room temperature with constant shaking. Subsequently, membranes were incubated overnight at 4°C with the specific primary antibody appropriately diluted in blocking solution (depending on specificities of antibodies by datasheet) (Table 1). Thereafter, membranes were incubated for 1 h at room temperature at constant agitation, with an HRP-conjugated secondary antibody suitably diluted (Table 1). Proteins detection was made using the Amersham ECL Select™ detection system (RPN2235; Cytiva Life Sciences, Marlborough, MA, USA) which highlights the HPR substrate by a chemiluminescent reaction. The membranes were

acquired using the ImageQuant™ LAS 4000 (GE Healthcare Life Sciences, Marlborough, MA, USA). A minimal contrast adjustment was applied to the images to enhance visibility, at the same time as ensuring that the integrity of the data was preserved.

The content of each protein investigated was assessed by determining the Brightness Area Product (BAP) of the protein band. Each target protein levels were then normalized to glyceraldehyde 3-phosphate dehydrogenase (GAPDH) content. Phosphorylated proteins were normalized by their respective total isoform. Ponceau S staining was used to normalize mono- and polyubiquitinated proteins and 3-nitrotyrosine content. Coomassie Blue staining (Coomassie Brilliant Blue G-250; #1610406; Bio-Rad Laboratories, Inc.) was used to normalize OXPHOS complexes. Data are expressed as the ratio between the BAP of target protein and that of housekeeping/total proteins (arbitrary units, AU).

## Carbonylated protein analysis

Frozen samples were homogenized in a lysis buffer (50 mM Tris-HCl, pH 7.6, 250 mM NaCl and 5 mM EDTA), supplemented with protease inhibitor 5X (Protease Inhibitor Cocktail; P8340) and phosphatase inhibitors 1X (P0044), both purchased from Sigma-Aldrich. Samples were incubated on ice for 20 min and centrifuged at 18,000 *g* for 20 min at 4°C. Protein concentration was determined using the RC DC™ protein assay kit (#5000122; Bio-Rad Laboratories, Inc.). Protein carbonylation levels were detected using the OxyBlot™ Kit (S7150; Millipore, Darmstadt, Germany) that provides reagents for sensitive immunodetection of carbonyl groups. Carbonyl groups in the protein side chains were derivatized to 2,4-dinitrophenylhydrazone (DNP) by reaction with 2,4-dinitrophenylhydrazine (DNPH). In detail, 10 μg of proteins for each muscle sample were denatured with 6% SDS (w/v). The DNPH solution was added to obtain the derivation; the reaction was stopped after 10 min of incubation at room temperature. The DNP-derivatized protein samples were separated by gradient precast protein gel electrophoresis (Any kD; #4569034; Bio-Rad Laboratories, Inc.). Proteins were transferred into nitrocellulose membranes (#1620113; Bio-Rad Laboratories, Inc.) at 100 V for 2 h at room temperature, and stained with Ponceau S (P3504; Sigma-Aldrich). The membranes were blocked with 3% BSA (w/v) in TTBS 1X for 1 h, then incubated with rabbit anti-DNP antibody overnight at 4°C and subsequently with a HRP-conjugated goat anti-rabbit IgG antibody. The positive bands were visualized as described before (Amersham ECL Select™ detection system; Cytiva Life Sciences). Total protein carbonylation

**Table 1. List of antibodies used for western blot evaluations.**

| Protein target | Antibody dilution | Antibody source | Company | Catalogue number | RRID |
|---|---|---|---|---|---|
| 3-Nitrotyrosine | 1:500 in milk 5% | Mouse monoclonal | Millipore | 05-233 | AB_11214452 |
| ACC | 1:1000 in BSA 5% | Rabbit monoclonal | Cell Signaling Technology | #3676 | AB_2219397 |
| p-ACC $_{(Ser79)}$ | 1:1000 in BSA 5% | Rabbit polyclonal | Cell Signaling Technology | #3661 | AB_330337 |
| Akt | 1:1000 in BSA 5% | Rabbit monoclonal | Cell Signaling Technology | #4685 | AB_2225340 |
| p-Akt $_{(Ser473)}$ | 1:1000 in BSA 5% | Rabbit polyclonal | Cell Signaling Technology | #9275 | AB_329825 |
| AMPKα | 1:1000 in BSA 5% | Rabbit polyclonal | Cell Signaling Technology | #2532 | AB_330331 |
| p-AMPKα $_{(Thr172)}$ | 1:1000 in BSA 5% | Rabbit monoclonal | Cell Signaling Technology | #4188 | AB_2169396 |
| Atg7 | 1:1000 in BSA 5% | Rabbit monoclonal | Cell Signaling Technology | #8558 | AB_10831194 |
| Catalase | 1:1000 in milk 5% | Rabbit polyclonal | Abcam | ab52477 | AB_868694 |
| Citrate synthase | 1:2000 in milk 5% | Rabbit polyclonal | Abcam | ab96600 | AB_10678258 |
| DRP1 | 1:1000 in BSA 5% | Rabbit monoclonal | Cell Signaling Technology | #8570 | AB_10950498 |
| p-DRP1 $_{(Ser616)}$ | 1:1000 in BSA 5% | Rabbit polyclonal | Cell Signaling Technology | #3455 | AB_2085352 |
| p-DRP1 $_{(Ser637)}$ | 1:1000 in BSA 5% | Rabbit polyclonal | Cell Signaling Technology | #4867 | AB_10622027 |
| GAPDH | 1:3000 in milk 5% | Rabbit monoclonal | Abcam | ab181603 | AB_2687666 |
| Glutathione peroxidase 1 | 1:1000 in milk 3% | Rabbit polyclonal | Abcam | ab22604 | AB_2112120 |
| LC3B II/I | 1:1000 in milk 5% | Rabbit polyclonal | Sigma-Aldrich | L7543 | AB_796155 |
| Mfn1 | 1:1000 in milk 5% | Mouse monoclonal | Abcam | ab57602 | AB_2142624 |
| Mfn2 | 1:1000 in milk 5% | Rabbit polyclonal | Abcam | ab50843 | AB_2235186 |
| Mono and polyubiquitinated proteins | 1:1000 in BSA 2% | Mouse monoclonal | Enzo Life Sciences | BML-PW8810-0500 | AB_2051891 |
| mTOR | 1:1000 in BSA 5% | Rabbit polyclonal | Cell Signaling Technology | #2972 | AB_330978 |
| p-mTOR $_{(Ser2448)}$ | 1:1000 in BSA 5% | Rabbit polyclonal | Cell Signaling Technology | #2971 | AB_330970 |
| MuSK | 1:1000 in milk 5% | Rabbit polyclonal | Abcam | ab92950 | AB_10712686 |
| p-MuSK $_{(Tyr755)}$ | 1:1000 in milk 5% | Rabbit polyclonal | Abcam | ab192583 | N/A |

*(Continued)*

**Table 1. (Continued)**

| Protein target | Antibody dilution | Antibody source | Company | Catalogue number | RRID |
|---|---|---|---|---|---|
| NF-H | 1:1000 in milk 5% | Mouse monoclonal | Cell Signaling Technology | #2836 | AB_10694081 |
| NF-L | 1:1000 in BSA 5% | Rabbit monoclonal | Cell Signaling Technology | #2837 | AB_823575 |
| nNOS | 1:1000 in milk 5% | Rabbit polyclonal | Abcam | ab3511 | AB_303860 |
| p-nNOS (Ser1417) | 1:1000 in milk 5% | Rabbit polyclonal | Abcam | ab5583 | AB_304964 |
| OPA1 | 1:3000 in milk 5% | Mouse monoclonal | Abcam | ab119685 | AB_10901464 |
| P70S6K | 1:1000 in BSA 5% | Rabbit polyclonal | Cell Signaling Technology | #34475 | AB_2943679 |
| p-P70S6K (Thr389) | 1:1000 in BSA 5% | Rabbit polyclonal | Cell Signaling Technology | #9205 | AB_330944 |
| Parkin | 1:1000 in milk 5% | Mouse monoclonal | Thermo Fisher Scientific | #39-0900 | AB_2533396 |
| PGC-1$\alpha$ | 1:1000 in milk 5% | Rabbit polyclonal | Abcam | ab54481 | AB_881987 |
| PINK1 | 1:500 in milk 5% | Rabbit polyclonal | Thermo Fisher Scientific | #PA1-16604 | AB_2164267 |
| S6 | 1:1000 in BSA 5% | Rabbit monoclonal | Cell Signaling Technology | #2217 | AB_331355 |
| p-S6 (Ser235/236) | 1:1000 in BSA 5% | Rabbit polyclonal | Cell Signaling Technology | #2211 | AB_331679 |
| Sialin/SLC17A5 | 1:1000 in milk 5% | Rabbit monoclonal | Abcam | ab314223 | N/A |
| SiRT1 | 1:1000 in BSA 5% | Mouse monoclonal | Cell Signaling Technology | #8469 | AB_10999470 |
| SOD1 | 1:1000 in milk 5% | Rabbit polyclonal | Abcam | ab16831 | AB_302535 |
| SQSTM1/p62 | 1:1000 in milk 5% | Rabbit polyclonal | Cell Signaling Technology | #5114 | AB_10624872 |
| TOM20 | 1:1000 in milk 5% | Rabbit polyclonal | Santa Cruz Biotechnology | Sc-11415 | AB_2207533 |
| Total OXPHOS rodent WB antibody cocktail | 1:1000 in milk 5% | Mouse monoclonal | Abcam | ab110413 | AB_2629281 |
| TTC11/Fis1 | 1:1000 in milk 5% | Rabbit polyclonal | Abcam | ab71498 | AB_1271360 |
| $\alpha$-mouse Ig/HRP | 1:5000 in milk 5% | Rabbit polyclonal | Agilent | P0260 | AB_2636929 |
| $\alpha$-rabbit IgG/HRP | 1:10,000 in milk 5% | Goat polyclonal | Cell Signaling Technology | #7074 | AB_2099233 |

Glyceraldehyde-3-phosphate dehydrogenase (GAPDH) is the housekeeping used for normalization.

**Table 2. List of primers used for real-time PCR evaluations.**

| Gene target | Forward 5′- to 3′ | Reverse 5′- to 3′ | NCBI transcript accession number |
|---|---|---|---|
| AChR$\alpha$ | GTAGAACACCCAGTGCTTCCA | GCCCGACCTGAGTAACTTCAT | NM_007389 |
| AChR$\gamma$ | CCTGTGGATATTGAGGGTGCC | CCAGTAGATACAACCGTCGGG | NM_009604 |
| Atrogin-1 | GCAAACACTGCCACATTCTCTC | CTTGAGGGGAAAGTGAGACG | NM_026346 |
| Gadd45$\alpha$ | GAAAGTCGCTACATGGATCAGT | AAACTTCAGTGCAATTTGGTTC | NM_007836 |
| GAPDH | CACCATCTTCCAGGAGCGAG | CCTTCTCCATGGTGGTGAAGAC | NM_001289726 |
| HDAC4 | CAGACAGCAAGCCCTCCTAC | AGACCTGTGGTGAACCTTGG | NM_207225 |
| LRP4 | GCACACGGAATAGCCAGCA | GGATACAGGTACATTCGCCAAG | NM_172668 |
| MuRF1 | ACCTGCTGGTGGAAAACATC | CTTCGTGTTCCTTGCACATC | NM_001039048 |
| MuSK | ATCACCACGCCTCTTGAAAC | TGTCTTCCACGCTCAGAATG | NM_001037127 |
| MyoG | GTGCCCAGTGAATGCAACTC | CGAGCAAATGATCTCCTGGGT | NM_031189 |
| NCAM1 | TGTTCAAGCAGACACACCGT | GGTTTCCACTCAGAGGCGAG | NM_001404722 |
| nNOS | GATGACAACCGGTACCACGA | GGCGGGAGGATCCAGTTAGG | NM_008712 |
| Nrf1 | GCACCTTTGGAGAATGTGGT | CTGAGCCTGGGTCATTTTGT | NM_001164227 |
| PGC-1$\alpha$ | ACCCCAGAGTCACCAAATGA | CGAAGCCTTGAAAGGGTTATC | NM_008904 |
| Rapsyn | ATATCGGGCCATGAGCCAGTAC | TCACAACACTCCATGGCACTGC | NM_001420848 |
| RUNX1 | CGGTAGAGGCAAGAGCTTCAC | CGGGCTTGGTCTGATCATCT | NM_001111023 |
| SiRT1 | CCGCGGATAGGTCCATATACT | AACAATCTGCCACAGCGTCA | NM_019812 |
| Tfam | GAGAGCTACACTGGGAAACCACA | CATCAAGGACATCTGAGGAAAA | NM_009360 |

Glyceraldehyde-3-phosphate dehydrogenase (GAPDH) is the housekeeping used for normalization.

levels were quantitatively analysed as described before by comparison of the signal intensity of immune-positive proteins normalized on total proteins amount (Ponceau S staining) and expressed as arbitrary units (AU).

### Gene expression analysis

Approximately 20 mg of pulverized gastrocnemius was used for RNA extraction with SV Total RNA Isolation System (Z3100; Promega Corp., Madison, WI, USA). Quantification of extracted RNA was then performed using a NanoDrop Instrument (Thermo Fisher Scientific) and 300 ng of each sample was reverse transcribed using the SuperScript$^{TM}$ III RT enzyme (18080093; Thermo Fisher Scientific). The cDNA thus obtained was analysed by quantitative real-time PCR (real-time PCR) (AB7500; Applied Biosystems, Foster City, CA, USA), using a SYBR® Green PCR Master Mix (4367659; Applied Biosystems) and the data were normalized to GAPDH content. Oligonucleotide primers were purchased from Sigma-Aldrich and are listed in Table 2. Differentially expressed genes were determined using a default threshold of 0.6. The difference between $C_t$ (cycle threshold) values was calculated for each mRNA by taking the mean $C_t$ of duplicate reactions and subtracting the mean $C_t$ of duplicate reactions for the reference RNA measured on an aliquot from the same reaction ($\Delta C_t = C_t$ target gene – $C_t$ reference gene). All samples were then normalized to the $\Delta C_t$ value of a calibrator sample to obtain a $\Delta\Delta C_t$ value ($\Delta C_t$ target – $\Delta C_t$ calibrator) by means of a comparative method.

### Statistical analysis

Data are represented as the mean ± SD, with individual values displayed. All dependent variables were analysed for normal distribution using the Shapiro–Wilk test. Group effects were evaluated by one-way analyses of variance (ANOVA). No significant group effects were observed for protein levels of p-Akt$_{(Ser473)}$, p-mTOR$_{(Ser2446)}$, NF-H, p-MuSK$_{(Tyr755)}$, SOD1, sirtuin1 (SiRT1), peroxisome proliferator-activated receptor-gamma coactivator-1$\alpha$ (PGC-1$\alpha$), p-5'-adenosine monophosphate-activated protein kinase (AMPK)$_{(Thr172)}$, p- acetyl-CoA carboxylase (ACC)$_{(Ser79)}$, fission 1 (Fis1), p-DRP1$_{(Ser616)}$, p-DRP1$_{(Ser637)}$, microtubule-associated protein 1A/1B-light chain 3 (LC3B II/I), PTEN induced kinase 1 (PINK1); mRNA levels of atrogin-1, muscle-specific receptor tyrosine kinase (MuSK), PGC-1$\alpha$, nuclear respiratory factor 1 (Nrf1); and mitochondrial respirometry values (OXPHOS and ETS). For statistically significant group effects, pairwise multiple comparison were performed by comparing means of each group to the mean of the reference group O, aiming to evaluate the effects of ageing and NO$_3^-$ supplementation (O *vs* Y and O *vs* ON). In detail, Dunnett's *post hoc* test or Brown–Forsythe ANOVA with Dunnett's T3 test, when variances between groups were different, were

performed. Differences in variance were assessed using the Brown–Forsythe test. Kruskal–Wallis tests followed by Dunn's multiple comparison tests were performed for non-parametric evaluations (protein levels of mono- and polyubiquitinated proteins, glutathione peroxidase (GPX), CS, ubiquitin-binding protein (p62); mRNA levels of muscle-specific ring finger protein-1 (MuRF1), growth arrest and DNA damage-45$\alpha$ (Gadd45$\alpha$), lipoprotein receptor-related protein 4 (LRP4), rapsyn, SiRT1, mitochondrial transcription factor A (Tfam); NOx concentration; NMJ parameters of fragmentation, overlap, endplate area, complexity). Two-way ANOVA was applied on mitochondrial complexes protein levels, with Dunnett's *post hoc* multiple comparison test. $P < 0.05$ was considered statistically significant. The statistical analyses were performed using Prism, version 8.4.2 (GraphPad Software Inc., San Diego, CA, USA).

## Results

### NO metabolism and $NO_3^-$ supplementation

The endogenous NO production system was assessed in Gas muscle by measuring expression levels of neuronal NOS (nNOS), the predominant isoform of the NOS family of enzymes in skeletal muscle. Transcript levels of nNOS did not differ among the three groups (Fig. 1*A*). However, the levels of phosphorylated nNOS at Ser1412 residue, which represents the active form of the enzyme, were significantly lower in O compared to Y ($P = 0.0346$) (Fig. 1*B*). The protein levels of Sialin, an anionic transporter mainly involved in active NO transport, were lower in O than in Y ($P = 0.0348$) (Fig. 1*C*). No significant differences were observed between ON and O.

Concentrations of $NO_3^-$ plus $NO_2^-$ (collectively referred to as NOx), determined in the soleus muscle, were not significantly different in O *vs* Y (Fig. 1*E*).

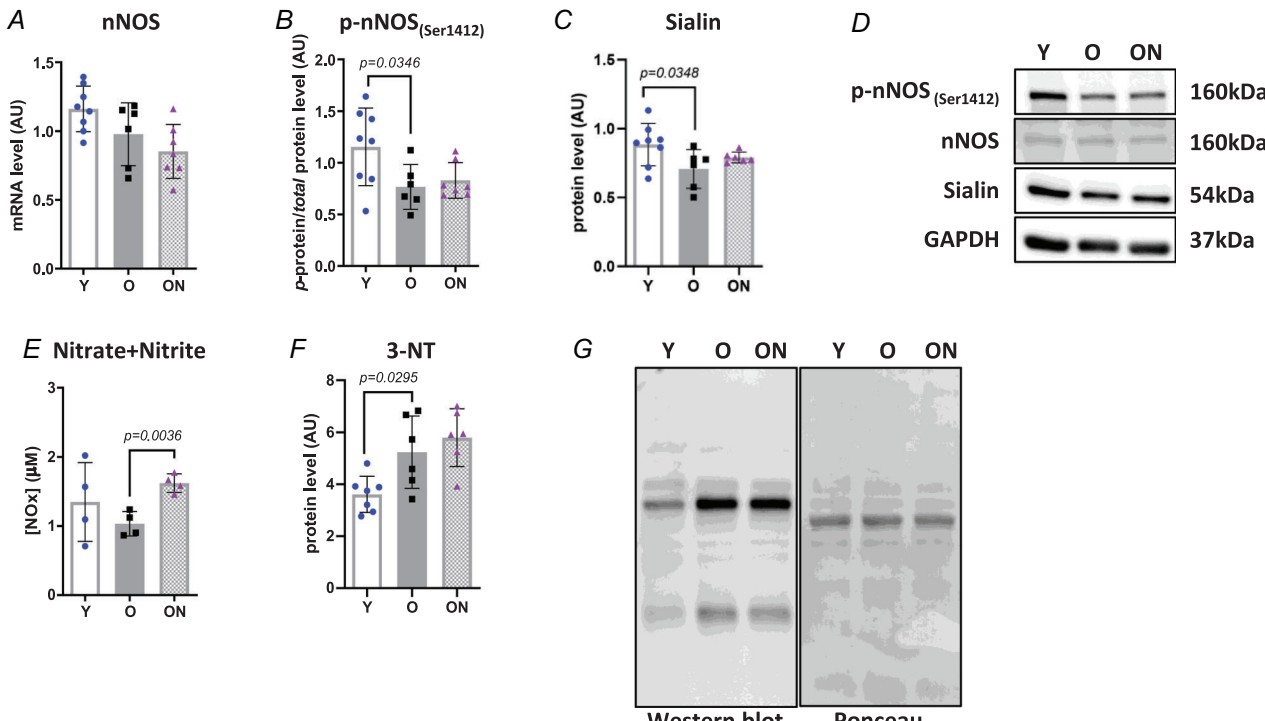

**Figure 1. Efficacy of $NO_3^-$ supplementation in old mice**
*A*, neuronal nitric oxide synthase (nNOS) mRNA expression determined by real-time PCR and normalized to glyceraldehyde-3-phosphate dehydrogenase (GAPDH) housekeeping gene. *B*, protein level of phosphorylated nNOS(Ser1412) normalized by the total nNOS form. *C*, protein levels of the transporter sialin. *D*, representative western blot images for all protein markers determined. *E*, $NO_3^-$ and $NO_2^-$ concentration determined by colorimetric assay (*n* = 4 per group). *F*, levels of 3-Nitrotyrosine (3-NT) protein modification. Protein content levels of all factors were determined by western blot and normalized to GAPDH level, except for 3-NT levels that were normalized against the total protein intensities measured from the same blot stained with Ponceau S (*G*). Except for $NO_3^-$ and $NO_2^-$ concentration determined in soleus muscle, all data were collected from gastrocnemius. Data are presented as the mean ± SD (*n* = 6–8 per group, unless otherwise indicated). $P < 0.05$ was considered statistically significant. Y = Young; O = Old; ON = Old + $NO_3^-$.

**Table 3. Descriptive data of mice from the three experimental groups.**

|  | Y | O | ON | *P* value |
|---|---|---|---|---|
| Age (months) | 7 | 24 | 24 | |
| *N* | 8 | 7 | 7 | |
| Body weight (gr) | 31 ± 1.92 | 35 ± 4.67 | 39.71 ± 5.08 | |
| Gastrocnemius (mg) | 169.5 ± 6.35 | 137.9 ± 11.64 | 142.8 ± 14.58 | Y *vs* O *P* < 0.0001 |
| Tibialis anterior (mg) | 55.48 ± 3.68 | 48.11 ± 5.15 | 49.27 ± 5.84 | Y *vs* O *P* = 0.0147 |
| Soleus (mg) | 10.44 ± 0.77 | 8.84 ± 0.81 | 9.60 ± 9.20 | Y *vs* O *P* = 0.0029 |
| EDL (mg) | 11.26 ± 0.91 | 9.98 ± 1.60 | 10.66 ± 0.85 | |
| *Muscle weight normalized on body weight* | | | | |
| Gastrocnemius | 5.48 ± 0.43 | 3.94 ± 0.25 | 3.62 ± 0.41 | Y *vs* O *P* < 0.0001 |
| Tibialis anterior | 1.79 ± 0.20 | 1.37 ± 0.16 | 1.24 ± 0.15 | Y *vs* O *P* < 0.0001 |
| Soleus | 0.33 ± 0.01 | 0.25 ± 0.01 | 0.24 ± 0.02 | Y *vs* O *P* = 0.0004 |
| EDL | 0.36 ± 0.03 | 0.28 ± 0.02 | 0.27 ± 0.03 | Y *vs* O *P* = 0.0003 |

Age (months), numerosity of the groups (*n*), body weight (gr), individual muscle weight of gastrocnemius, tibialis anterior, soleus and EDL (mg), and ratio of individual muscle weight on body weight. Data are presented as the mean ± SD. $P < 0.05$ was considered statistically significant. Y = Young; O = Old; ON = Old + $NO_3^-$.

However, NOx concentration was higher in ON than in O ($P = 0.0036$) (Fig. 1*E*), suggesting efficacy of $NO_3^-$ treatment in enhancing the bioavailability of NO. Muscle levels of 3-NT (Fig. 1*F*), a product of tyrosine nitration mediated by ROS, were determined in Gas. They were higher in O than in Y ($P = 0.0295$), suggesting an age-induced nitrosative stress. By contrast, 3-NT levels in ON mice were not significantly higher than those in O, suggesting that $NO_3^-$ supplementation did not enhance nitrosative stress in ageing.

### Muscle phenotype

O mice had a higher body weight compared to Y (Table 3). No difference was observed between O and ON. Gas, TA and soleus muscles had lower weights in O compared to Y (Table 3). No differences were observed between muscle weights of O and ON.

The CSA of muscle fibres in Gas was determined from transversal cryo-sections stained with an anti-dystrophin antibody. Mean CSA of muscle fibres was lower in O compared to Y ($P = 0.0015$), whereas it was greater in ON compared to O ($P = 0.0485$) (Fig. 2*A*). The number of muscle fibres counted in the whole Gas cross-section was also lower in O than in Y mice ($P = 0.0436$) (Fig. 2*B*), with no changes observed in the ON group.

To evaluate whether non-contractile tissue infiltration had replaced muscle tissue, we assessed fibrotic tissue deposition using Picrosirius Red staining. The percentage of the area occupied by fibrous tissue was higher in O than in Y ($P = 0.0067$) (Fig. 2*C*). In ON mice, the latter parameter was lower than in O ($P = 0.0112$) and did not differ from Y (Fig. 2*D*).

To assess whether the relative distribution of type 1 (slow) and type 2 (fast) fibres was affected by ageing, cross cryo-sections were stained with an antibody against the slow MHC isoform. As expected, slow fibres were few and localized to a small area of the Gas. The percentage of type 1 fibres was greater in O compared to Y ($P = 0.0088$), suggesting the expected shift in favour of an oxidative phenotype (Fig. 2*D*). No differences were seen following $NO_3^-$ supplementation. Consistently, the percentage of fast type MHC showed the opposite trend, being lower in O compared to Y mice ($P = 0.0043$). No significant change was observed in ON (Fig. 2*E*).

The percentage of central nuclei within the Gas muscle fibres was higher in O compared to Y ($P = 0.0168$). No difference was found between ON and O mice (Fig. 2*F*).

### Intracellular pathways controlling muscle mass

Expression levels of MuRF1 and atrogin-1, key ubiquitin ligases involved in the ubiquitin-proteasome pathway, were determined. MuRF1 was upregulated in O *vs* Y Gas muscle ($P = 0.0124$) (Fig. 3*A*). Consistently, levels of polyubiquitinated proteins were higher in O *vs* Y ($P = 0.0442$) (Fig. 3*C*). No differences were found in expression of MuRF1 and atrogin-1, or in the content of polyubiquitinated proteins, between ON and O mice (Fig. 3*A–C*).

Protein levels of phosphorylated Akt and mTOR, major kinases of the IGF-1/Akt/mTOR pathway controlling muscle protein synthesis, were not significantly different in O *vs* Y (Fig. 3*D* and *E*). Similarly, phosphorylation levels of the major downstream kinases of Akt and mTOR, namely S6 and its kinase P70S6K, showed no difference between O and Y (Fig. 3*F* and *G*). Following $NO_3^-$

supplementation, whereas phosphorylation levels of Akt and mTOR were not different between ON *vs* O and Y, S6 and P70S6K had significantly higher phosphorylation in ON than in O mice ($P = 0.0324$ and $P = 0.0145$, respectively) (Fig. 3*F* and *G*).

## NMJ stability

Fragmentation of the post-synaptic side of NMJ was significantly greater in O *vs* Y mice ($P < 0.001$) (Fig. 4*A*), whereas overlap was notably lower ($P = 0.0003$) (Fig. 4*B*). Endplate area was larger ($P = 0.0018$), and compactness

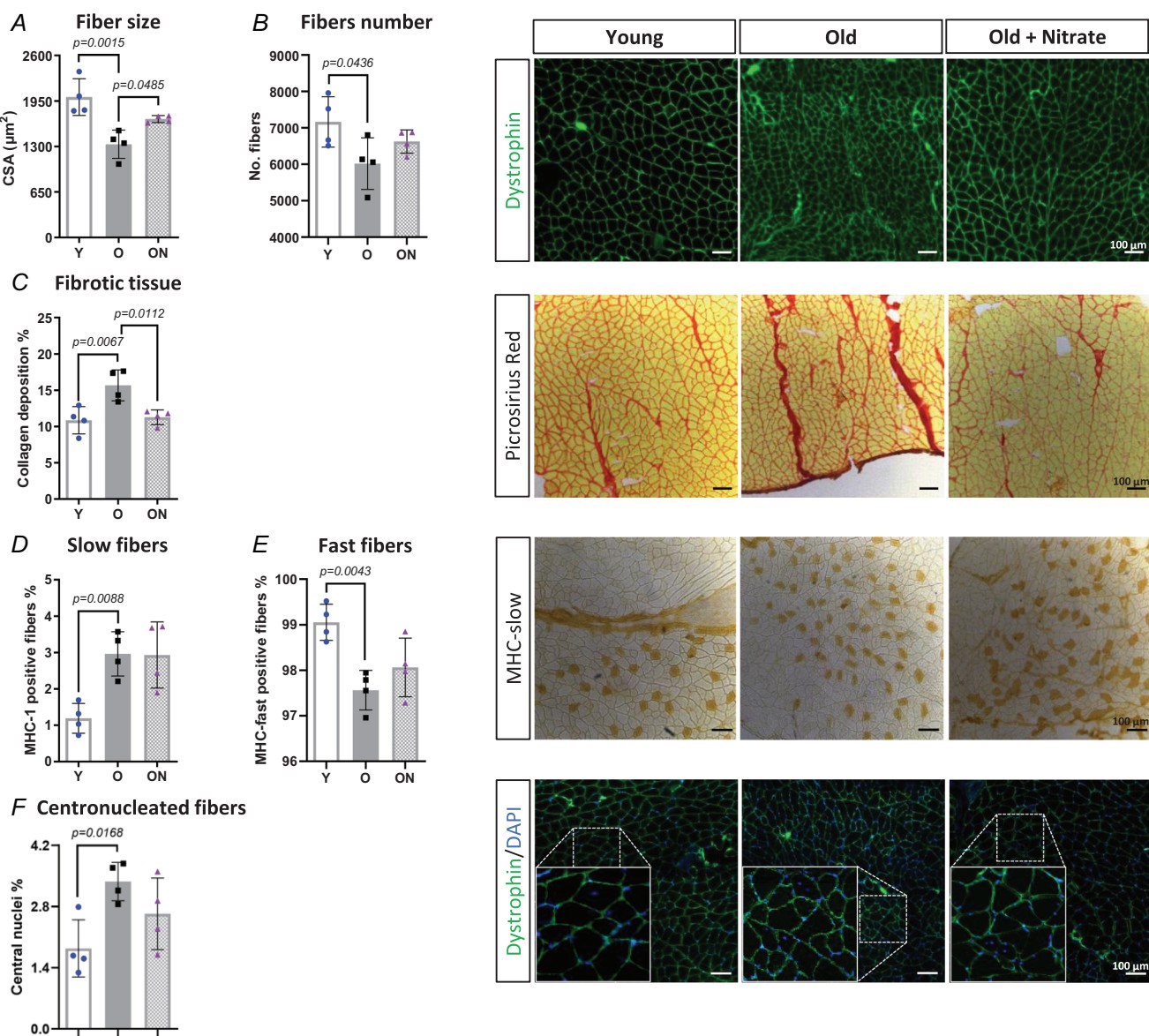

**Figure 2. $NO_3^-$ supplementation ameliorates ageing muscle phenotype in old mice**
*A*, quantification of muscle fibres cross-sectional area (CSA) and (*B*) number of fibres determined by immunofluorescence on muscle cross-sections stained with anti-dystrophin antibody as shown in representative images on the right (scale bar = 100 μm). *C*, quantification of collagen deposition by Picrosirius Red staining on muscle cross-sections as shown in the representative images on the right (scale bar = 100 μm). *D* to *E*, quantification of slow and fast fibres by immunohistochemistry on muscle cross-sections stained with myosin heavy chain (MHC) slow and MHC fast antibodies. Representative images stained by anti-MHC slow antibody are shown on the right (scale bar = 100 μm). *F*, quantification of centronucleated fibres determined by immunofluorescence on muscle cross-sections stained with dystrophin (green) and DAPI (blue). Representative images are shown on the right (scale bar = 100 μm). All data were collected from gastrocnemius muscle. Data are presented as the mean ± SD (*n* = 4 per group). $P < 0.05$ was considered statistically significant. Y = Young; O = Old; ON = Old + $NO_3^-$.

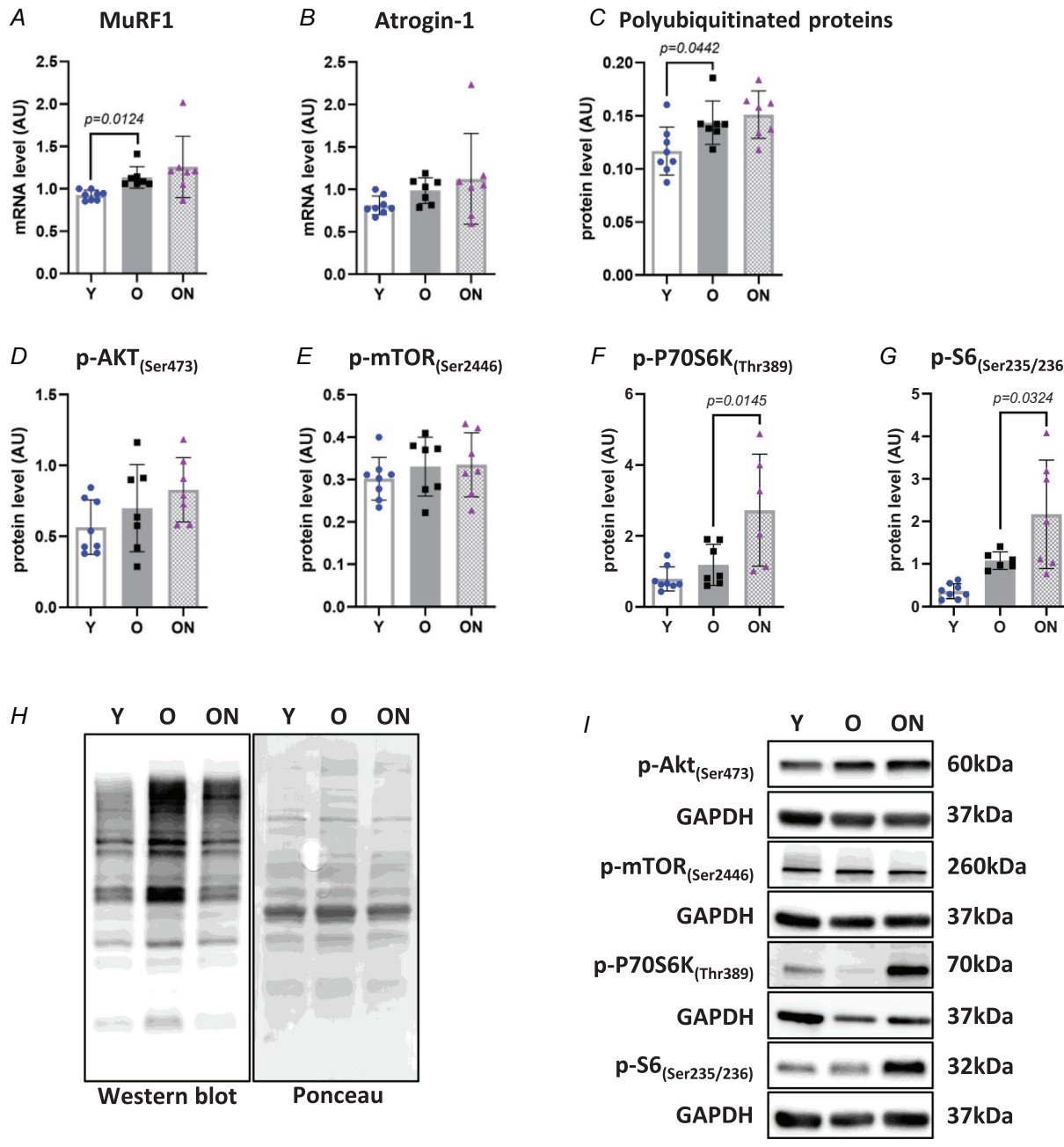

**Figure 3. NO$_3^-$ supplementation enhances downstream factors of the Akt/mTOR pathway in old mice**
*A* and *B*, quantification of muscle-specific ring finger protein-1 (MuRF1) and atrogin-1 mRNA expression by real-time PCR. Expression data were normalized to glyceraldehyde-3-phosphate dehydrogenase (GAPDH). *C*, levels of polyubiquitinated proteins determined by western blot. The intensity of total polyubiquitinated proteins was normalized against the total protein intensities measured from the same blot stained with Ponceau S. Representative images are shown in (*H*). *D* to *G*, determination of phosphoprotein levels of AKT$_{(Ser473)}$, mTOR$_{(Ser2446)}$, P70S6K$_{(Thr389)}$ and S6$_{(Ser235/236)}$ by western blot. Protein levels were normalized to GAPDH. *I*, representative western blot images of the Akt/mTOR pathway. All data were collected from gastrocnemius muscle. Data are presented as the mean ± SD (*n* = 6–8 per group). *P* < 0.05 was considered statistically significant. Y = Young; O = Old; ON = Old + NO$_3^-$.

was reduced ($P = 0.0262$) in O than in Y (Fig. 4C and *D*). Complexity on presynaptic side was also higher in O compared to Y ($P = 0.0416$) (Fig. 4*E*). Interestingly, in ON, fragmentation was lower ($P = 0.0036$) and overlap was higher ($P = 0.0003$) compared to O, and undistinguishable from Y (Fig. 4*A* and *B*). Endplate area was lower ($P = 0.0478$) (Fig. 4*C*) and compactness was greater ($P = 0.0001$) in ON compared to O mice (Fig. 4*D*).

Expression levels of factors potentially involved in denervation-reinnervation cycles were quantified. O showed higher levels of Gadd45$\alpha$ ($P = 0.0031$), myogenin-G (MyoG) ($P = 0.0196$), runt-related transcription factor 1 (RUNX1) (0.0014), acetylcholine receptor subunit $\gamma$ (AChR$\gamma$) ($P = 0.0232$) and NCAM1 ($P = 0.0288$) compared to Y mice (Fig. 5*A*), suggesting an increased denervation rate in O mice relative to Y. NO$_3^-$ supplementation did not modulate the transcript levels of any of such factors in ON compared to O (Fig. 5*A*). NCAM1 transcript levels showed a trend to be lower in ON than in O. Interestingly, percentage of fibres positive for NCAM1 expression was higher in O compared to Y ($P < 0.0001$) and lower in ON compared to O ($P = 0.0002$) (Fig. 5*B*). Neurofilament light (NF-L), which is primarily represented in axonal structures, was lower in O *vs* Y ($P = 0.0015$) (Fig. 5*D*), whereas no significant recovery was detected in ON *vs* O. No changes in neurofilament

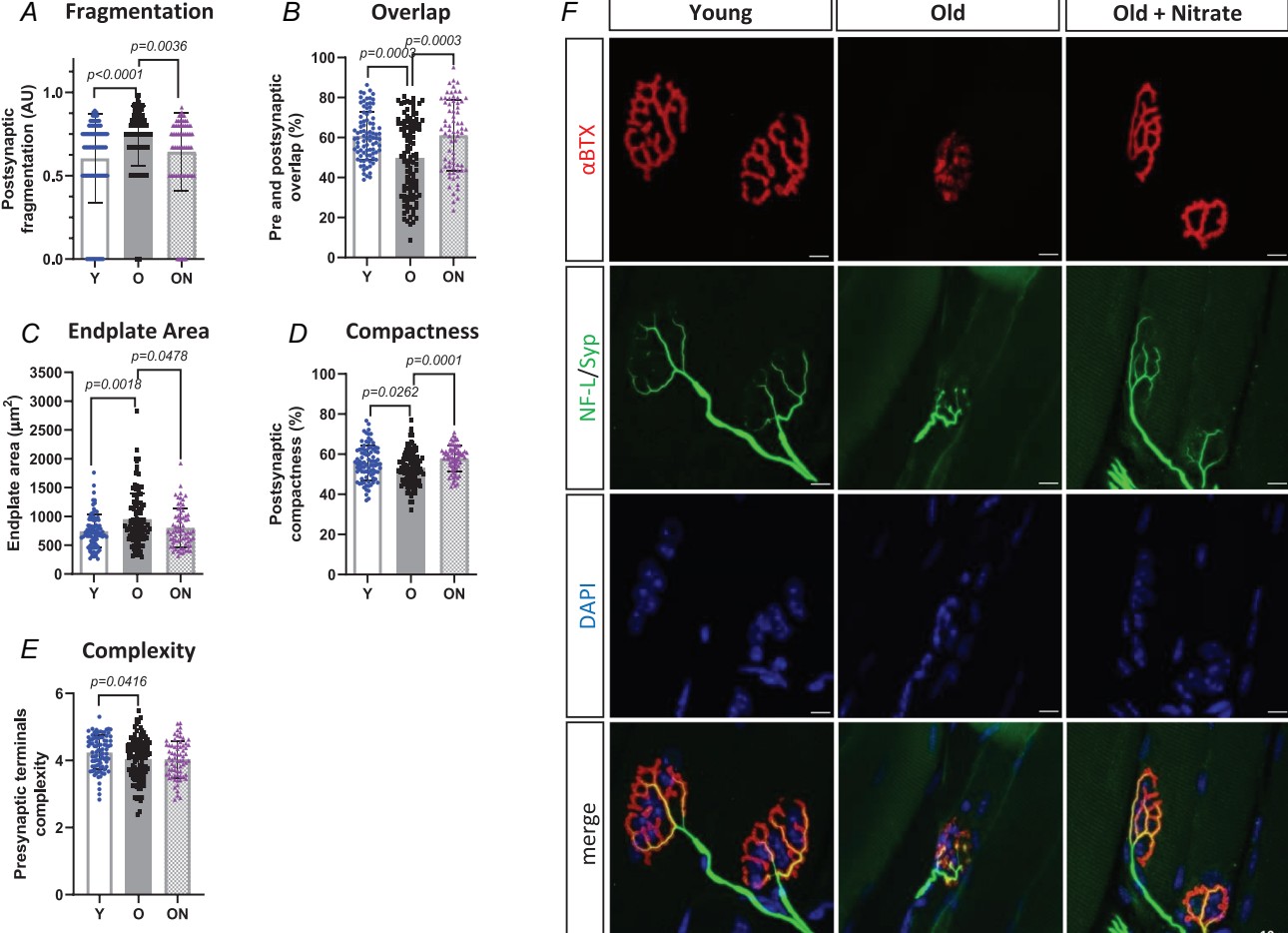

**Figure 4. NO$_3^-$ supplementation positively remodels the morphology of the neuromuscular junction (NMJ) in old mice**
*A*, index of postsynaptic fragmentation of the NMJ, as a result of the function (1 − 1/number of AChR clusters). *B*, percentage of overlap between the presynaptic terminal and the postsynaptic side of the NMJ. *C*, endplate area of the postsynaptic side of the NMJ (expressed in µm$^2$). *D*, percentage of compactness of the postsynaptic structure. *E*, branching complexity of the presynaptic terminal, as a result of the function log$_{10}$(number of terminal branches × number of branch points × total length of branches). *F*, immunofluorescence images of longitudinal extensor digitorum longus (EDL) whole-mounted muscles stained with $\alpha$-bungarotoxin ($\alpha$-BTX) (postsynaptic side, in red), neurofilament light (NF-L) and synaptophysin (Syp, presynaptic side, in green), DAPI (nuclei, in blue) and merge of all channels. Scale bar = 10 µm. Data are presented as the mean ± SD ($n$ = 70–110 NMJ analysed per group; three animals each group). $P < 0.05$ was considered statistically significant. Y = Young; O = Old; ON = Old + NO$_3^-$.

heavy (NF-H) protein levels were observed across any of the groups.

The mRNA levels of several factors involved in the stability and maintenance of a clustered and functional AChR structure, namely LRP4, rapsyn and MuSK, and the content of p-MuSK$_{(Tyr755)}$ did not change under any of the conditions studied (Fig. 6*A* and *B*). However, acetylcholine receptor subunit $\alpha$ (AChR$\alpha$) levels were significantly higher in O mice compared to Y ($P = 0.0100$) (Fig. 6*A*), consistent with the morphological observations indicating a wider but disorganized post-synaptic structure.

## Redox state

Oxidative stress could play an important role in the ageing process, contributing to the accumulation of RONS and giving rise to muscle and NMJ damage.

Carbonylated proteins were significantly higher in Gas of O than in Gas of Y ($P = 0.0156$) (Fig. 7*A*). Consistently, higher levels of 3-NT (Fig. 1*F*) and lower levels of GPX ($P = 0.0363$) (Fig. 7*D*), a major redox buffer, were observed in O *vs* Y. By contrast, levels of two other major redox buffers, SOD1 and catalase, remained unchanged between O and Y (Fig. 7*D*). To directly assess rate of mitochondrial hydrogen peroxide (mtH$_2$O$_2$) production, we performed high-resolution respirometry analysis on

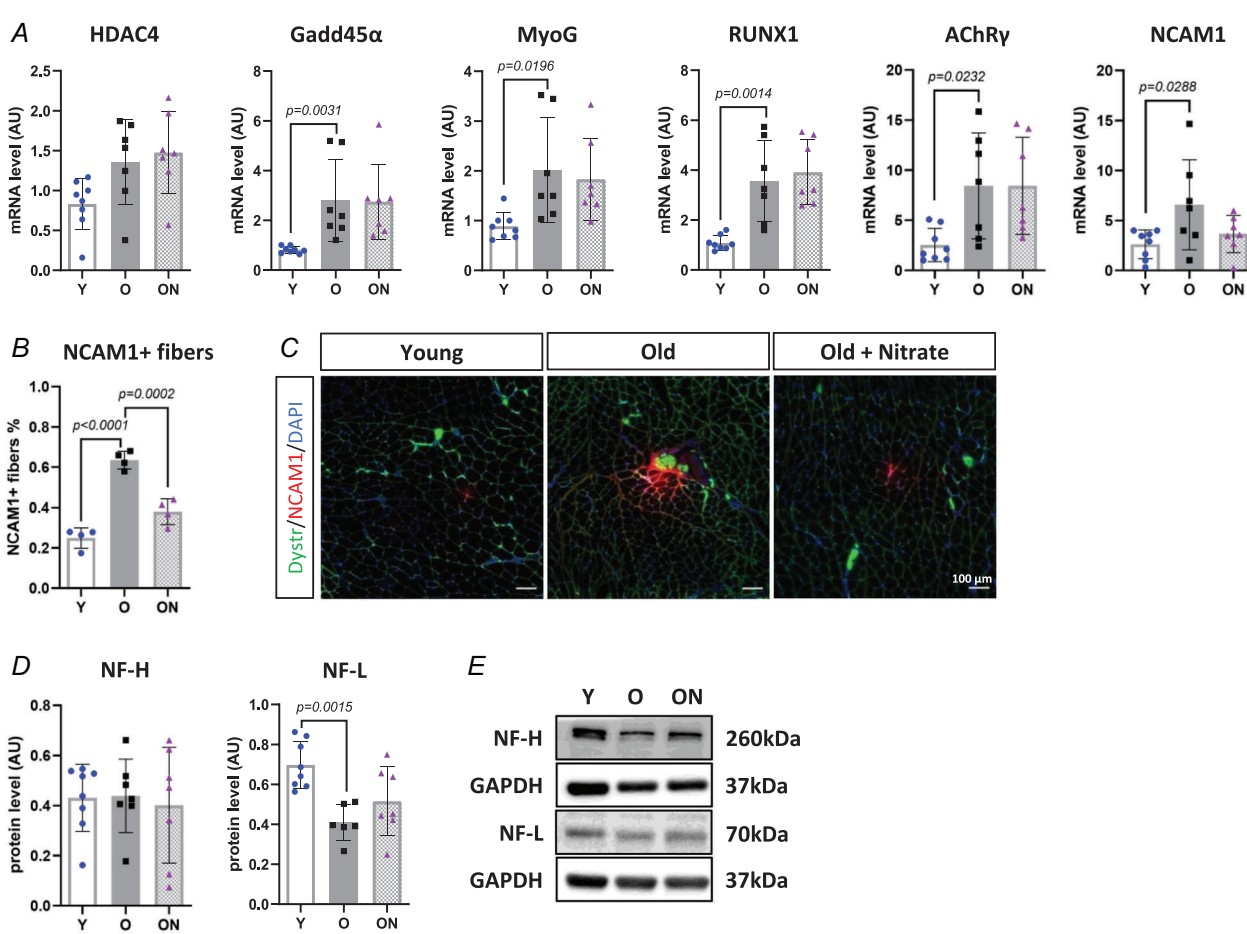

**Figure 5. Age-related denervation-associated factors are partially ameliorated by NO$_3^-$ supplementation**

*A*, mRNA quantification of denervation-associated factors histone deacetylase-4 (HDAC4), growth arrest and DNA damage-45$\alpha$ (Gadd45$\alpha$), myogenin-G (MyoG), runt-related transcription factor 1 (RUNX1), acetylcholine receptor subunit $\gamma$ (AChR$\gamma$) and neural cell adhesion molecule 1 (NCAM1) by real-time PCR. Expression data were normalized to glyceraldehyde-3-phosphate dehydrogenase (GAPDH). *B*, percentage of NCAM1 positive fibres (*n* = 4 per group) by immunofluorescence. Representative images of muscle cross-sections stained with dystrophin (green), NCAM1 (red) and DAPI (blue) are shown in (*C*), scale bar = 100 µm. *D*, quantification of protein levels of neurofilament heavy (NF-H) and neurofilament light (NF-L) by western blot. Protein levels were normalized to GAPDH. *E*, representative western blot of NF-H and NF-L. All data were collected from gastrocnemius muscle. Data are presented as the mean ± SD (*n* = 6–8 per group, unless otherwise indicated). *P* < 0.05 was considered statistically significant. Y = Young; O = Old; ON = Old + NO$_3^-$.

permeabilized muscle fibres from TA muscle. Emission of $mtH_2O_2$ did not differ between O and Y.

$NO_3^-$ supplementation had a relevant impact on redox status. Carbonylated protein content was lower in ON *vs* O ($P = 0.0226$), returning to levels undistinguishable from those in Y. Importantly, $mtH_2O_2$ production was lower in ON than in O ($P = 0.0499$) and GPX levels were higher in ON compared to O ($P = 0.0425$) and also undistinguishable from Y.

## Mitochondrial markers and function

To assess mitochondrial mass, we quantified protein levels of translocase of outer mitochondrial membrane 20 (TOM20), an outer mitochondrial membrane protein that is often used as mitochondrial mass marker (Fig. 8*A*). CS, a major mitochondrial enzyme (Fig. 8*B*), and the content of enzymes in the respiratory chain were also determined (Fig. 8*C*). O mice showed higher levels of

TOM20 ($P = 0.0489$), CS ($P = 0.0453$), and Complexes I (CI) and IV (CIV) ($P = 0.0263$ and $P < 0.0001$, respectively) in comparison to Y. Content in respiratory chain CIV was also higher in ON ($P = 0.0188$) compared to O. No differences were found for respiratory chain CII, CIII and CV among groups.

Mitochondrial biogenesis was assessed based on the expression of SiRT1, PGC-1$\alpha$, Nrf1 and Tfam, as well as on protein content of SiRT1 and PGC-1$\alpha$ (Fig. 8). No differences were found between O and Y. However, SiRT1 and Tfam were higher in ON compared to O ($P = 0.0472$ and $P = 0.0023$, respectively). No differences were found for PGC-1$\alpha$ and Nrf1 between ON and O.

Mitochondrial dynamics were assessed by the analysis of pro-fission and pro-fusion protein markers. Among pro-fusion proteins, mitofusin 2 (Mfn2) content was higher in O than in Y ($P = 0.0244$) (Fig. 9*A*), whereas no differences were observed for mitofusin 1 (Mfn1) and OPA1 (Fig. 9*A*). Among pro-fission proteins, none were significantly altered (Fig. 9*B*).

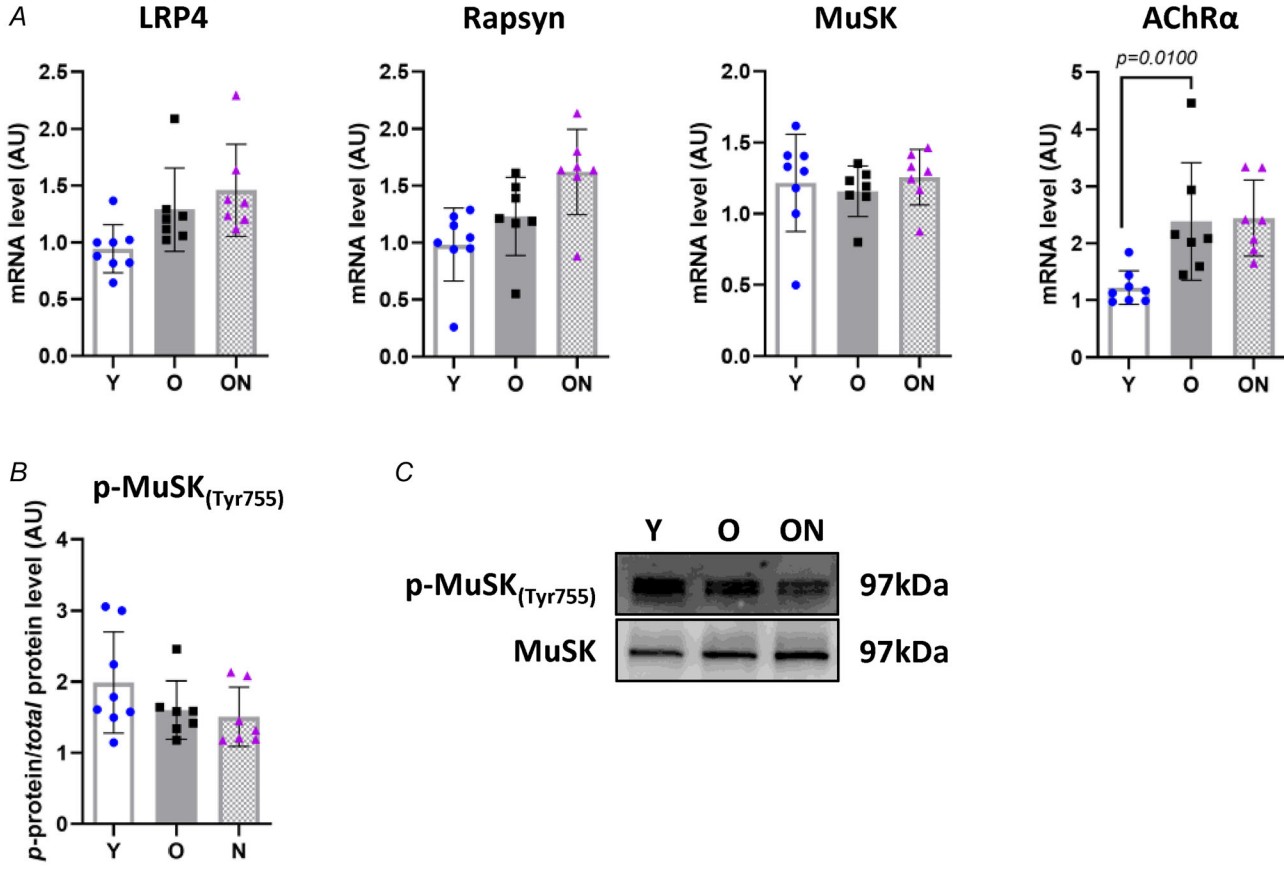

**Figure 6. Factors involved in NMJ stability are unaffected by ageing and $NO_3^-$ supplementation**
*A*, mRNA quantification of low-density lipoprotein receptor-related protein 4 (LRP4), rapsyn, muscle-specific receptor tyrosine kinase (MuSK) and acetylcholine receptor subunit $\alpha$ (AChR$\alpha$) by real-time PCR. Expression data were normalized to glyceraldehyde-3-phosphate dehydrogenase (GAPDH). *B*, phosphorylation level of MuSK(Tyr755) expressed through the ratio between the phosphorylated and the total MuSK form. Representative western blots are shown in (*C*). All data were collected from gastrocnemius muscle. Data are presented as the mean $\pm$ SD ($n = 7$–8 per group). $P < 0.05$ was considered statistically significant. Y = Young; O = Old; ON = Old + $NO_3^-$.

Among markers of autophagy and mitophagy, autophagy-related 7 (Atg7) (Fig. 9*C*) and Parkin (Fig. 9*D*) were significantly upregulated in O compared to Y ($P = 0.0117$ and $P = 0.0476$, respectively). LC3B II/LC3B I ratio, p62 and PINK1 levels were unchanged, with no differences observed between ON and O.

Mitochondrial function was determined in permeabilized skeletal muscle fibres by *ex vivo* high-resolution respirometry. Fig. 10 shows that leak respiration (LEAK), maximal ADP-stimulated mitochondrial respiration (OXPHOS) supported by CI and CII, ETS capacity and OXPHOS coupling efficiency were not different in O *vs* Y. However, OXPHOS coupling efficiency was higher in ON *vs* O ($P = 0.0118$). Consistent with lack of functional deterioration, levels of p-AMPK$_{(Thr172)}$ and p-ACC$_{(Ser79)}$, which are markers

of cellular energy balance, were unchanged under all conditions (Fig. 8*L*).

## Discussion

Sarcopenia has a major impact on human health. A better understanding of the underlying molecular mechanisms and the finding of compounds that could mitigate sarcopenia is a critical scientific challenge. We hypothesized that $NO_3^-$ supplementation positively affects NMJ stability in old mice by modulating factors related to redox balance, mitochondria and muscle protein synthesis, or to all of them.

Our findings indicate that: (1) 24-month-old mice had a clear sarcopenic phenotype, with altered NMJ

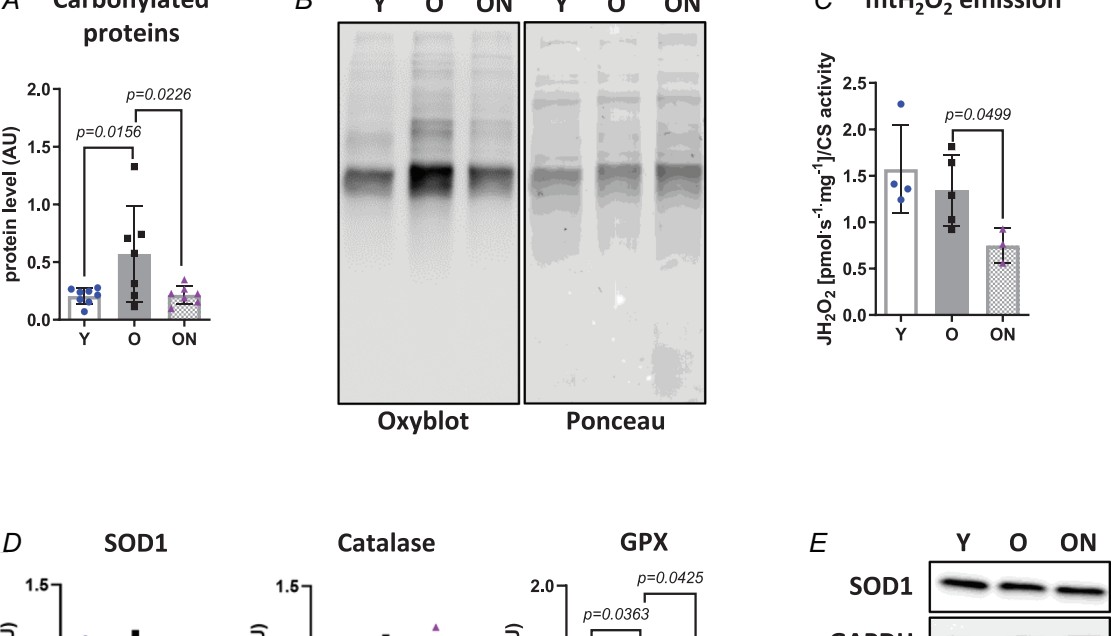

**Figure 7. $NO_3^-$ supplementation ameliorates the redox status of old mice**
*A*, quantification of carbonylated protein levels by OxyBlot. The intensity of OxyBlot bands was normalized against the total protein intensities measured from the same blot stained with Ponceau S. Representative images are shown in (*B*). *C*, quantification of mitochondrial $H_2O_2$ emission by O2k-Fluorometer. Data are normalized to citrate synthase activity ($n = 3–5$ per group). *D*, protein levels of superoxide dismutase 1 (SOD1), catalase and glutathione peroxidase (GPX) antioxidant enzymes determined by western blot. Protein levels were normalized to glyceraldehyde-3-phosphate dehydrogenase (GAPDH). Representative western blots are shown in (*E*). Except for $H_2O_2$ emission determined in tibialis anterior muscle, all data were collected from gastrocnemius. Data are presented as the mean $\pm$ SD ($n = 6–8$ per group, unless otherwise indicated). $P < 0.05$ was considered statistically significant. Y = Young; O = Old; ON = Old + $NO_3^-$.

morphology and redox unbalance, but showed no signs of mitochondrial alteration or enhanced mitochondrial $H_2O_2$ production; (2) 2 months of $NO_3^-$ supplementation increased NO availability, blunted muscle fibre atrophy and fibrous infiltration, restored NMJ morphology,

mitigated redox unbalance, and enhanced the activity of key components of the Akt/mTOR pathway. Notably, $NO_3^-$ supplementation also resulted in some relevant effects on mitochondria; namely, lower $H_2O_2$ production and improved OXPHOS coupling efficiency.

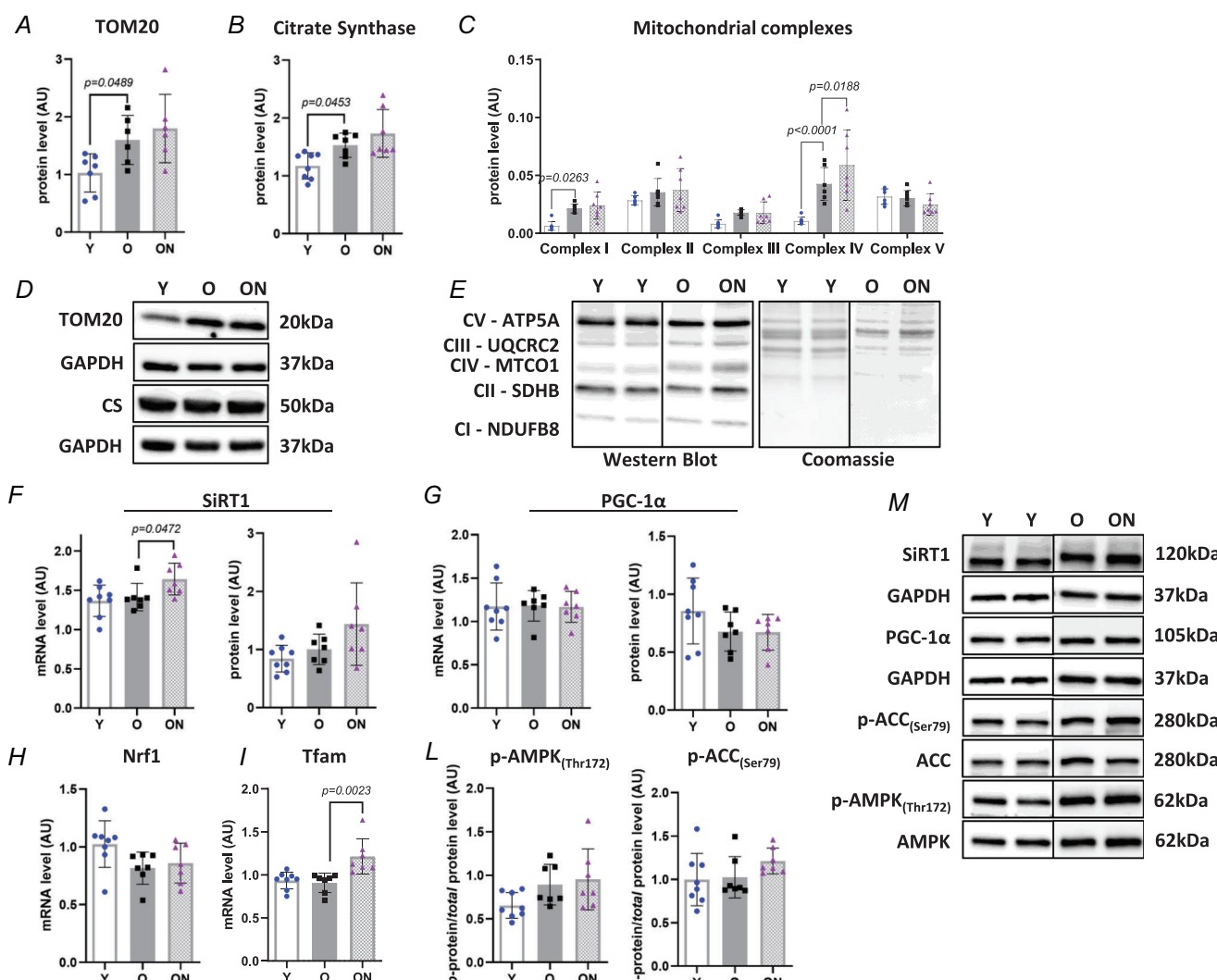

**Figure 8. Mild impact of ageing and $NO_3^-$ supplementation on mitochondrial markers**
Key mitochondrial enzymes and markers involved in mitochondrial mass and biogenesis were analysed. Quantification of protein levels of (*A*) translocase of outer mitochondrial membrane 20 (TOM20), (*B*) citrate synthase (CS) and (*C*) mitochondrial complexes assessed by western blot. *D-E*, representative western blots for TOM20, CS and mitochondrial complexes. *F*, mRNA and protein levels of sirtuin1 (SiRT1) determined with real-time PCR and western blot, respectively. *G*, mRNA and protein levels of peroxisome proliferator-activated receptor-gamma coactivator-1α (PGC-1α) determined with real-time PCR and western blot, respectively. *H*, mRNA level of nuclear respiratory factor 1 (Nrf1) and (*I*) mRNA level of mitochondrial transcription factor A (Tfam) determined by real-time PCR. *L*, activation level of 5′-adenosine monophosphate-activated protein kinase (AMPK) and its downstream acetyl-CoA carboxylase (ACC) determined by western blot through the ratio between the content in their phosphorylated ($AMPK_{(Thr172)}$, $ACC_{(Ser79)}$) and total form (AMPK and ACC, respectively). *M*, representative western blots for all protein markers determined. Protein expression levels were normalized to glyceraldehyde-3-phosphate dehydrogenase (GAPDH) except for proteins intensity of mitochondrial complexes normalized to protein loading using Coomassie blue staining. mRNA expression levels were normalized to GAPDH. All data were collected from gastrocnemius muscle. Data are presented as the mean ± SD (*n* = 6–8 per group). $P < 0.05$ was considered statistically significant. Y = Young; O = Old; ON = Old + $NO_3^-$.

### Ageing profile of 24-month-old mice

Consistent with previous findings in mice (Ratto et al., 2019), the 24-month-old mice (O) used in the present study were demonstrated to be a good model of ageing.

**Muscle structure.** The 24-month-old mice showed typical ageing phenotype features with altered muscle quantity and quality: loss of weight (Table 3), lower muscle fibres CSA, lower number of Gas fibres, fast to slow muscle fibre type shift and larger area occupied by fibrous tissue

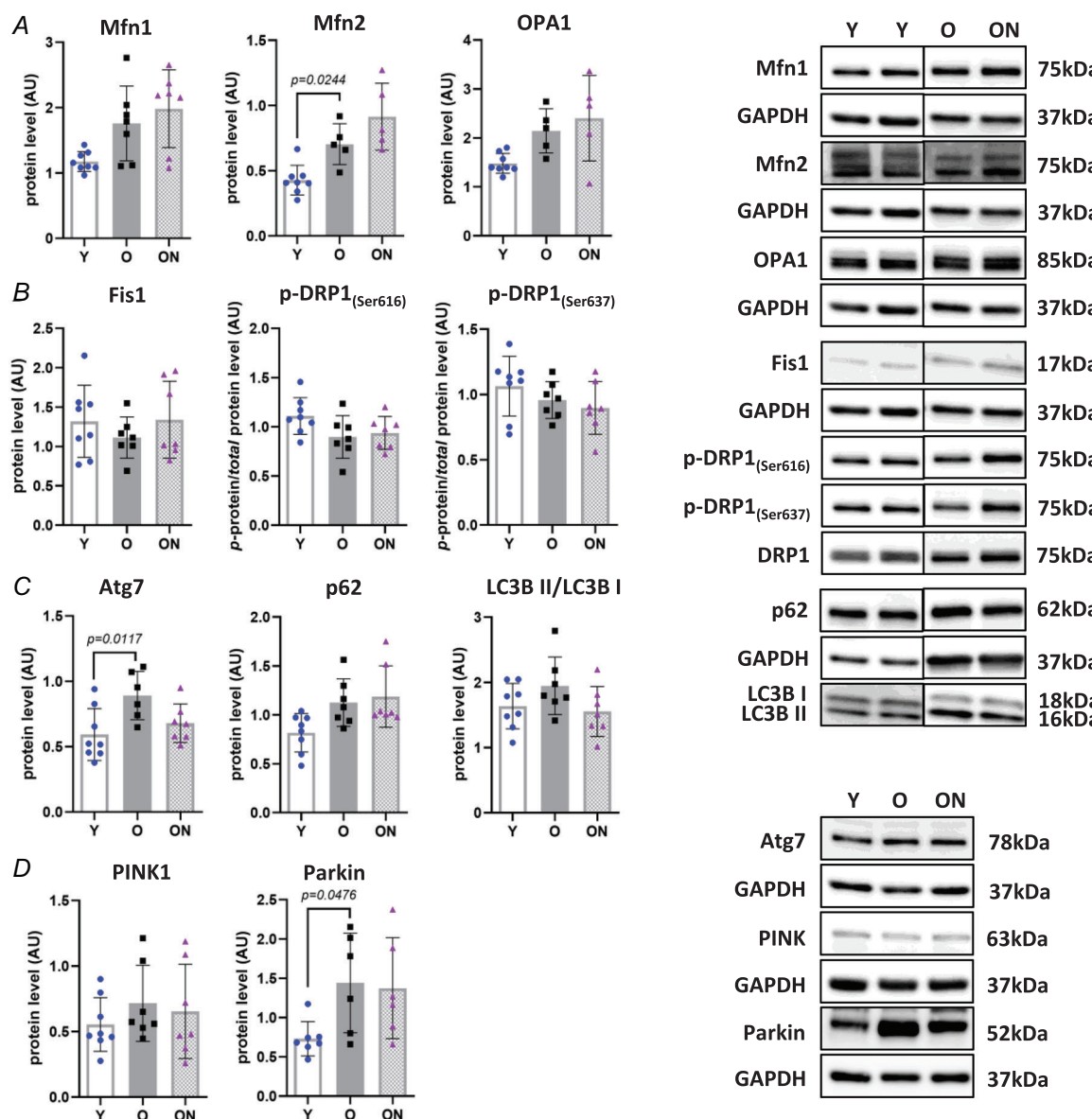

**Figure 9. Age-related changes in mitochondrial dynamics and mitophagy are not affected by $NO_3^-$ supplementation**

*A*, protein levels of mitochondrial fusion markers: mitofusin 1 (Mfn1), mitofusin 2 (Mfn2) and OPA1. *B*, protein levels of mitochondrial fission markers: fission 1 (Fis1) and phosphorylated levels of $DRP1_{(Ser616)}$ and $DRP1_{(Ser637)}$, both normalized to the total form of dynamin-related protein 1 (DRP1). *C*, protein levels of autophagy-related 7 (Atg7), ubiquitin-binding protein (p62) and microtubule-associated protein 1A/1B-light chain 3 (LC3B II), with the latter normalized to the LC3B I form. *D*, protein levels of mitophagy-related factors PTEN induced kinase 1 (PINK1) and Parkin. All analyses were performed by western blot and normalized to glyceraldehyde-3-phosphate dehydrogenase (GAPDH), unless otherwise indicated. Representative western blot images of all markers are presented on the right. All data were collected from gastrocnemius muscle. Data are presented as the mean ± SD (*n* = 5–8 per group). Comparison with *P* < 0.05 was considered statistically significant. Y = Young; O = Old; ON = Old + $NO_3^-$.

(Fig. 2). Additionally, old muscles showed upregulation of MuRF1, an ubiquitin ligase widely used as marker for the activation of the ubiquitin-proteasome pathway controlling muscle protein breakdown, along with an increase in polyubiquitinated proteins (Fig. 3). The latter results, together with unchanged levels of key kinases in the Akt/mTOR pathway (Akt, mTOR, S6 and P70S6) (Fig. 3), which modulate muscle protein synthesis, are consistent with muscle fibre atrophy. However, enhanced atrogenes expression probably does not play a major role in sarcopenia because atrogenes were found mostly unchanged (Sandri et al., 2013) or even down-regulated (Edstrom et al., 2006) in ageing. Furthermore, because no measurements of proteasome activity were performed, it may be possible that the observed increase in poly-ubiquitinated proteins is a consequence of ineffective proteasome function.

**NMJ.** NMJ integrity was assessed through morphological analysis (Jones et al., 2016) complemented by analysis of expression or content of proteins potentially involved in denervation or NMJ stability. Although the relationship between NMJ morphological and functional alterations and sarcopenia is complex and not fully understood (Jones et al., 2016; Willadt et al., 2016; Willadt et al., 2018), NMJ instability is widely considered a major factor in denervation and reinnervation phenomena underlying muscle ageing (Anagnostou & Hepple, 2020; Hepple & Rice, 2016).

Consistent with previous findings in ageing rodents, morphology of the NMJ was altered in O mice compared to Y, showing significantly higher fragmentation, a clear index of age-related instability and dysfunction in rodents (Cheng et al., 2013; Rudolf et al., 2014), lower pre/post-synaptic overlap, larger endplate area and lower compactness (Fig. 4). Such NMJ alterations were supported by upregulation of denervation-associated factors (Gadd45$\alpha$, MyoG, RUNX1, AChR$\gamma$ and NCAM1) (Hepple & Rice, 2016) in O mice compared to Y (Fig. 5), consistent with previous findings (Aare et al., 2016; Soendenbroe et al., 2020). Furthermore, enhanced expression of NCAM1 and a higher percentage of NCAM1 positive fibres (Fig. 5) in O suggest a higher rate of denervation (Hendrickse et al., 2018; Soendenbroe et al., 2019). Indeed, selective expression of NCAM1 in partially denervated muscles was suggested to promote and enhance the functional expansion of the synaptic territory, stimulating a structural reorganization of motoneurons to facilitate reinnervation (Chipman et al., 2014). Expression levels of several factors involved in NMJ structural stability (LRP4, rapsyn and MuSK) did not show any age-related perturbations (Fig. 6) (Zhao et al., 2018), despite the observed negative remodelling of the morphology. Anyhow, consistent with such morphological alterations, AChR$\alpha$ mRNA levels were higher in O mice, indicating a non-compacted and disorganized NMJ structure, probably as a result of reduced interaction and overlap with the nervous component (Brown et al., 2019).

**Reactive oxygen and nitrogen species (RONS) balance.**
Oxidative and nitrosative stress is generally defined as an unbalance between RONS and antioxidant systems. Uncontrolled oxidative stress was shown to be detrimental for skeletal muscle homeostasis, causing muscle atrophy and NMJ instability (Dobrowolny et al., 2018). Moreover, higher oxidation has been observed in skeletal muscle of old mice (Palomero et al., 2013).

A higher content in carbonylated proteins (Fig. 7) and higher 3-NT levels (Fig. 1), which are known markers of redox unbalance (Beltran Valls et al., 2015), suggest prominent irreversible protein modifications driven by RONS. Such accumulation is probably enhanced by a compromised scavenging capacity of antioxidant

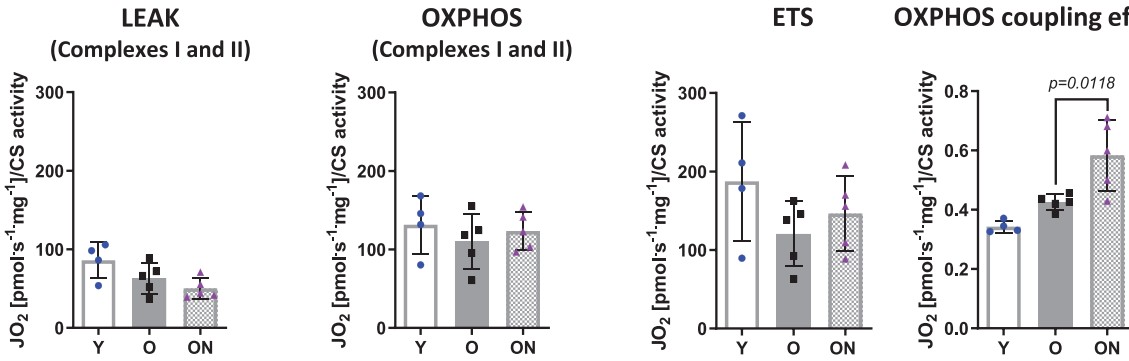

**Figure 10. Mitochondrial respiration is maintained in ageing and its efficiency is enhanced by NO$_3^-$ supplementation**
Mitochondrial respiration variables obtained by high-resolution respirometry in permeabilized tibialis tnterior fibres: LEAK respiration, OXPHOS capacity, electron transport system (ETS) capacity and OXPHOS coupling efficiency [(OXPHOS-LEAK)/OXPHOS]. Data were normalized to citrate synthase activity. Data are presented as the mean ± SD ($n$ = 4–5 per group). $P < 0.05$ was considered statistically significant. Y = Young; O = Old; ON = Old + NO$_3^-$.

enzymes, as manifested both by significantly lower protein levels of GPX, a critical enzyme that reduces $H_2O_2$ preventing oxidative damage, and by lack of response of SOD1 and catalase, for which protein levels remained unchanged compared to Y. The latter interpretation is further supported by unaltered mitochondrial $H_2O_2$ emission in O mice (Fig. 7). Therefore, it can be hypothesized that oxidative stress in O mice is primarily the result of a compromised antioxidant defence system rather than an increase in RONS production itself.

**Mitochondrial markers and function.** The classical free radical theory of ageing, based on primary mitochondrial abnormalities leading to redox unbalance and mitochondrial DNA mutations, is widely questioned (Larsson et al., 2019). Larger mitochondrial networks are considered to have enhanced function and relatively lower ROS production, whereas mitochondria undergoing fission are associated with lower function, relatively higher ROS production and a propensity for mitophagy (Romanello & Sandri, 2010). In ageing, contradictory results have been reported, showing both larger and more complex mitochondrial networks (Leduc-Gaudet et al., 2015), as well as smaller and less complex ones (Huang et al., 2010). However, mitochondria are still considered key players in the ageing process, being very abundant on both sides of the NMJ (Rygiel et al., 2016), where their ROS production could drive NMJ instability.

The observed higher mitochondrial fusion, indicated by higher Mfn2 levels (with trends towards higher levels of Mfn1 and OPA1, $P = 0.0509$ and $P = 0.0505$, respectively) (Fig. 9) occurred alongside unchanged mitochondrial biogenesis (Fig. 8) and fission (Fig. 9). The upregulation of key mitochondrial enzymes, such as CS and components of the electron transport chain (Fig. 8), along with the lack of significant changes in cellular energy balance sensors (Fig. 8), suggests a compensatory response to mitochondrial damage associated with ageing. Importantly, mitochondrial $H_2O_2$ emission did not differ between Y and O mice (Fig. 7). The available data on markers of mitochondrial biogenesis and dynamics in ageing humans are limited and somewhat contradictory. Some studies report reduced CS activity, PGC-1$\alpha$ mRNA and NRF1/Tfam protein levels in older individuals (Crane et al., 2010; Ghosh et al., 2011), whereas others show preserved mitochondrial markers (Johannsen et al., 2012; Wyckelsma et al., 2017) and elevated PGC-1$\alpha$ protein levels (Johannsen et al., 2012). Regarding mitochondrial dynamics, transcript levels of Mfn2 and DRP1 were shown to decrease with age, although their protein expression was found to increase, potentially as a protective mechanism. Such findings are consistent with those observed in our animal model.

Furthermore, markers of mitophagy and autophagy suggest that mitophagy may not be effective in old mice.

For mitophagy to be effective, damaged mitochondria must be separated from the healthy mitochondrial network through fission. Once separated, they can be targeted by the mitophagy machinery (i.e. PINK1 and Parkin). The accumulation of PINK1 initiates a process that leads to the translocation of Parkin, which is typically cytosolic and inactive, to the mitochondria. This results in the degradation of the damaged mitochondria within autophagosomes. Therefore, an increase in Parkin alone (Fig. 9) may not guarantee effective activation of mitophagy in O mice. Moreover, the unchanged levels of the LC3B II/LC3B I ratio (Fig. 9), an indicator of autophagosome formation, may suggest that the mechanisms responsible for the removal of damaged mitochondria are not enhanced, despite higher Atg7 (Fig. 9). The latter suggestion aligns with impairment of quality control systems observed in animal models of precocious ageing because of autophagy inhibition (Carnio et al., 2014).

Collectively, these findings do not support mitochondrial alterations in 24-month-old mice. Consequently, the observed redox unbalance (Fig. 7), potentially causing NMJ instability (Figs 4, 5), probably does not originate from mitochondrial dysfunction, at least in mice. This suggestion is consistent with the controversial role of mitochondria in ageing that has emerged subsequent to the classical theory of ageing was questioned (Larsson et al., 2019). Indeed, NMJ alterations are among the earliest phenomena of ageing, potentially preceding muscle atrophy (Deschenes et al., 2010) and possibly mitochondrial alterations, which could be a consequence rather than a cause of NMJ instability.

## $NO_3^-$ supplementation ameliorates muscle phenotype in old mice

NO in our body primarily originates from two sources: dietary intake of $NO_3^-$ and $NO_2^-$ (mainly from green leafy vegetables) (i.e. the nitrate-nitrite-NO pathway) and NO synthase-derived catabolism of L-arginine (i.e. the L-arginine-NOS-NO pathway). Circulating $NO_3^-$ is actively taken up by the salivary glands and secreted in saliva. In the mouth, $NO_3^-$ is reduced to the more reactive $NO_2^-$ anion by commensal bacteria. In blood and tissues, $NO_2^-$ is further metabolized into several bioactive nitrogen oxides, including NO (Lundberg et al., 2018). NO, a volatile and unstable compound, is involved in various molecular processes (Lundberg & Govoni, 2004). During ageing, one or both NO production mechanisms may be impaired, leading to lower NO bioavailability (Di Massimo et al., 2006; Donato et al., 2018).

We assessed the efficiency of endogenous NO production system based on expression levels and phosphorylation of nNOS (Fig. 1), which has been shown to activate the enzyme in several tissues (Chakrabarti

et al., 2012; Mount et al., 2006; Zhao et al., 2021). Lower phosphorylation of nNOS in O mice suggests impaired enzyme activity. We focused on nNOS because of its functional relevance in skeletal muscle and its structural role. Both nNOS and endothelial NOS isoforms are expressed in muscle tissue. However, nNOS is considered the main source of NO production, whereas endothelial NOS, although constitutively present in skeletal muscle, is mainly expressed in cells that share an endothelial/epithelial identity (Tengan et al., 2012). Moreover, nNOS has been shown to actively interact with dystrophin at the level of the sarcolemma (Tengan et al., 2012).

Protein levels of sialin, the anion transporter involved in $NO_3^-$ translocation into the cell (Qin et al., 2012), were much lower in O mice (Fig. 1), depicting an impaired transport system in skeletal muscle. Skeletal muscle serves as a $NO_3^-$ storage (Park et al., 2021). The significantly higher levels of NOx metabolites in ON mice (Fig. 1) indicate that $NO_3^-$ supplementation was effective in reaching the muscles in old mice. No difference was observed between Y and O, possibly as a result of high variability in the Y group (Justice et al., 2015). Unchanged levels of 3-NT, representing the NO-induced modification of tyrosine residues of proteins used as a marker of nitrosative damage (Herce-Pagliai et al., 1998), suggest no toxicity of $NO_3^-$ supplementation in old mice.

$NO_3^-$ administration did not affect body weight or the weight of individual hindlimb muscles (Table 3). However, it did improve muscle quality by increasing the CSA of muscle fibres and decreasing fibrotic infiltration (Fig. 2). A probable explanation for the unchanged muscle weight despite the higher CSA is that the lower fibrotic infiltration may compensate for the higher muscle fibres mass. Interestingly, the trend toward a lower percentage of centrally nucleated fibres in ON mice (Fig. 2) may be driven by a NO-dependent tissue repair (Filippin et al., 2009).

Overall, 2 months of $NO_3^-$ supplementation can increase nitrate/nitrite levels in skeletal muscle of 24-month-old mice and ameliorate phenotype in terms of muscle quantity and quality (i.e. higher CSA and lower fibrotic infiltration).

## $NO_3^-$ supplementation promotes NMJ remodelling in old mice

The substantial fragmentation of the post-synaptic side observed in O mice underwent a significant amelioration in old animals treated with $NO_3^-$ (Fig. 4). The NO-induced improvement of the NMJ is supported by larger pre/post-synaptic overlap, lower endplate area and higher compactness (Fig. 4) in ON mice compared to O. Such a finding is consistent with the lower percentage of NCAM1 positive fibres in ON (Fig. 5). The latter finding is consistent with the observations that NO treatment

is able to improve motor function and motility in both murine (Justice et al., 2015) and human (Coggan et al., 2020) models of ageing.

No changes in the expression levels of several denervation-associated markers, which are potentially affected by ageing, were observed following $NO_3^-$ supplementation (Fig. 5). One potential explanation for this discrepancy could be that, notwithstanding the overall improvement in NMJ morphology, the underlying stress signals may not be fully eliminated by the treatment, thereby maintaining the elevated expression of such markers. This hypothesis is supported by the observation that 3-NT levels were unchanged in mice treated with $NO_3^-$ (Fig. 1). Furthermore, it may be hypothesized that additional nerve-dependent mechanisms contribute to the observed improvement in NMJ morphology in ON mice. Indeed, NO plays a relevant role in the regeneration and plasticity of axons (Nishiyama et al., 2003). Following nerve injury, Schwann cells play a role in guiding axonal regeneration, driven by sprouting factors released by muscle fibres (Tam & Gordon, 2003). The involvement of NO in the reorganization of nerve terminals and solicitation of terminal Schwann cells was observed after 14 days of nerve crush (Marques et al., 2006).

In general, NMJ morphology shows a positive remodelling in old mice following 2 months of supplementation with inorganic $NO_3^-$, resembling Y. Such amelioration does not appear to depend on several structural proteins involved in NMJ assembly, suggesting that alternative molecular mechanisms may be involved.

## Mechanisms mediating the effects of $NO_3^-$ supplementation

The synthetic pathway was found to be enhanced by NO-donor treatments both *in vitro* (Wang et al., 2018) and *in vivo* (Petrick et al., 2025; Wang et al., 2022). Anabolic pathways, orchestrated by Akt/mTOR, were shown to be critical for proper innervation and NMJ stability (Baraldo et al., 2020; Castets et al., 2020), with some proteins also playing a structural role. Interestingly, lower mTORC1 signalling was shown to lead to NMJ fragmentation in mice (Baraldo et al., 2020). Furthermore, the inhibition of protein synthesis in mouse muscle fibres was shown to be sufficient to affect NMJ stability and its presynaptic component (McCann et al., 2007). Interestingly, synaptic deconstruction and axonal withdrawal were found to correlate with magnitude of protein synthesis inhibition. It was hypothesized that such effect was a consequence of the blockage of the synthesis of potential factors involved in the retrograde signalling pathway, which is known to be part of the mechanisms for maintaining synaptic connections and which requires continuous replenishment (Gromova & La Spada, 2020). In this scenario, it is conceivable that an improvement

of the anabolic pathway may mitigate NMJ deterioration. Accordingly, following 2 months of $NO_3^-$ treatment, we show a significant increase in the phosphorylation levels of downstream factors of the mTOR pathway (i.e. S6 and its kinase, P70S6K), despite no changes in mTOR protein levels or its phosphorylation (Fig. 3). The downstream factors of mTOR are actively involved in the translation machinery recruited to synthesize new proteins (Magnuson et al., 2012). The strong induction of the synthetic pathway by $NO_3^-$ supplementation could also play a role in the higher CSA of muscle fibres found after 2 months of treatment of ON mice (Fig. 2).

$NO_3^-$ supplementation had a profound effect on mitigating redox balance in old mice. The reduction in carbonylated protein content, in parallel with the increase in GPX protein levels (Fig. 7) in ON mice to levels similar to Y, indicates that $NO_3^-$ supplementation can boost antioxidant defences, thereby limiting oxidative stress damage. These findings are consistent with the direct activation of antioxidant genes by NO in a rodent model of oxidative stress induced by hypoxia (Singh et al., 2012). In addition to this, consistent with the literature (Brunetta et al., 2022), $NO_3^-$ supplementation resulted in a significant improvement in mitochondrial OXPHOS coupling efficiency (OXPHOS-LEAK)/OXPHOS) (Fig. 10) compared to O, primarily because of reduced LEAK respiration and unchanged OXPHOS. The lower LEAK respiration indicates a lower protonmotive force, which correlates with lower mitochondrial ROS levels, because mitochondrial $H_2O_2$ is sensitive to changes in protonmotive force. Indeed, ON mice showed a reduction in the intrinsic production of mitochondrial $H_2O_2$ (Fig. 7). Notably, although $NO_3^-$ supplementation can inhibit cytochrome oxidase (Brown & Cooper, 1994), the determination of coupling efficiency reported here was probably not affected by such a phenomenon. Indeed, cytochrome oxidase, responsible for transferring electrons from CIII to CIV, is not observed in the LEAK state because only CI and CII are stimulated. Similarly, this effect is not detected in OXPHOS. Interestingly, lower LEAK respiration and higher mitochondrial efficiency following nitrate supplementation were also observed in mitochondria from human vastus lateralis muscle (Larsen et al., 2011).

Therefore, the results reported here suggest that $NO_3^-$ supplementation may directly or indirectly modulate mitochondrial function, possibly reducing electron leakage (Sarti et al., 2012) or enhancing mitochondrial efficiency, thereby lowering ROS production. Notably, consistent with the notion that SiRT1 is required for the beneficial effects of dietary $NO_3^-$ on mitochondria (Brunetta et al., 2022), SiRT1 expression levels were found to be higher in old mice supplemented with $NO_3^-$ (Fig. 8). Because carbonylation is an irreversible protein modification induced by ROS, it can be hypothesized

that during the 2 months of $NO_3^-$ supplementation, the rate of protein carbonylation decreased, allowing protein catabolic systems (e.g. ubiquitin-proteasome system and autophagy) to effectively eliminate them, thereby reducing their concentration in the muscles of ON mice.

Overall, the present findings suggest that both the enhancement of Akt/mTOR pathways and the blunting of ROS unbalance could play a role in NO-induced amelioration of NMJ morphology and, in turn, muscle phenotype.

Importantly, nitrate supplementation could ameliorate sarcopenia through several mechanisms, including improved vascular function and enhanced muscle oxygenation, both of which have been observed in animals and humans (Bailey et al., 2015; Ferguson et al., 2013a, b). Indeed, cellular energy and oxygen levels are both regulators of the synthetic pathway (Saxton & Sabatini, 2017). Therefore, nitrate supplementation could enhance protein synthesis by optimizing the cellular environment and providing the energy required to activate the anabolic pathway.

Interestingly, improved vascular control and skeletal muscle $O_2$ delivery during exercise is predominant in fast-twitch type II muscles, where the reduction of $NO_2^-$ to NO is potentiated by low $O_2$ environments (Ferguson et al., 2013b). Because muscle phenotype is slower in humans than in mice, the nitrate-induced enhancement of $O_2$ delivery is expected to be less marked in humans compared to mice. However, there is strong evidence that nitrate supplementation enhances submaximal oxygen uptake, muscle oxygenation and exercise tolerance in humans, suggesting that the slower muscle fibres phenotype does not significantly hinder the efficacy of nitrate on muscle itself (Lansley, Winyard, Bailey et al., 2011; Lansley, Winyard, Fulford et al., 2011; Larsen et al., 2007).

Finally, because NO can influence satellite cells activation, the rescue of satellite cells regenerative capacity could also contribute to muscle mass maintenance following NO administration (Leiter et al., 2012). Indeed, the capacity of satellite cells to respond to stretch and exercise stimuli is known to be impaired in age-related muscle atrophy. Notably, nitrate administration was able to enhance vascular endothelial growth factor expression in aged mice, suggesting a role for improved vascularization in this process (Leiter et al., 2012).

## Limitations of the present study

To achieve our goal, we had to perform a large number of analyses. The significant amount of tissue required, relative to the size of mice muscles and the number of animals used (around eight per group), along with practical and ethical reasons, prevented us from conducting all analyses on the same muscles.

Consequently, NMJ morphology was studied in the EDL muscles; muscle mass, phenotype, intracellular pathways and biomarkers were analysed in the Gas; *ex vivo* mitochondrial function in the TA; and NOx concentrations in the Soleus. This choice probably does not affect the general message of the study for several reasons. In mice, the EDL, Gas and TA are all predominantly fast muscles (with up to ∼90% fast muscle fibres). The EDL is particularly suited for NMJ morphological analysis, whereas the Gas size enabled us to perform most of the analyses (immunohistochemical and molecular), except for *ex vivo* mitochondrial function.

A limitation of the present study could be the small sample size for *ex vivo* mitochondrial function determinations, especially in the Y group. Such limited numerosity increases the risk of type II errors, which could result in the failure to detect true effects. However, we mitigated this risk by testing each sample twice, averaging the results and performing experiments on the same day. Future studies should aim for larger sample sizes to enhance the reliability of the results.

Finally, the lack of data on whether nitrate supplementation ameliorated muscle function limits the enthusiasm for its potential impact as a treatment to counteract sarcopenia.

## Conclusions

The findings of the present study indicate that 2 months of $NO_3^-$ supplementation can alleviate sarcopenia in mice by blunting NMJ deterioration and denervation, enhancing the activity of the Akt/mTOR pathway, and counteracting redox unbalance by increasing GPX content and reducing mitochondrial $H_2O_2$ production. Importantly, a detailed analysis of mitochondrial mass, biogenesis, dynamics and function shows no significant alterations with age. These findings suggest that, in mice, sarcopenia-related NMJ alterations may be an early phenomenon, whereas mitochondrial dysfunction may develop over time, possibly because of NMJ alterations. Notably, dietary intake of $NO_3^-$-rich foods, such as leafy greens and beets, was shown to significantly increase NO levels, representing a promising nutritional approach to address sarcopenia and supporting beneficial effects on cardiovascular and muscular health in humans (Hoon et al., 2015; Larsen et al., 2011; Sim et al., 2021).

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

## Additional information

### Data availability statement

The data that support the findings of this study are partially available in the supporting information for the review process and fully available from the corresponding author upon reasonable request.

## Competing interests

The authors declare that they have no competing interests.

## Author contributions

Experiments were performed in the Laboratory of Muscle Plasticity of the Department of Molecular Medicine, University of Pavia, Pavia, Italy. M.A.P and R.B conceived the study. M.A.P obtained the financial support. M.R, L.Z, L.B, B.G, S.P, R.B and M.A.P contributed to the experimental design. P.R provided the animals. M.R, L.Z, L.B, C.S and C.G collected and analysed the data. M.R, R.B and M.A.P wrote the original draft. All authors critically revised the draft and approved the final version of the manuscript submitted for publication.

## Funding

The present work was funded by the PRIN project 'NeuAge' (2017CBF8NJ_001) to MAP.

## Acknowledgements

We thank the staff members of the animal facility 'Centro di servizio per la gestione unificata delle attività di stabulazione e di radiobiologia' of the University of Pavia, Pavia, Italy, for hosting the animals; the OPBA of the University of Pavia for support for drawing up the animal protocol; and Patrizia Vaghi and Amanda Oldani of 'Centro Grandi Strumenti', University of Pavia, Pavia, Italy, for their technical support and assistance with the experimental setup for confocal microscopy experiments.

Open access publishing facilitated by Universita degli Studi di Pavia, as part of the Wiley - CRUI-CARE agreement.

## Keywords

ageing, extensor digitorum longus, gastrocnemius, mitochondria, neuromuscular junction, nitrate supplementation, oxidative stress, tibialis anterior

## Supporting information

Additional supporting information can be found online in the Supporting Information section at the end of the HTML view of the article. Supporting information files available:

**Peer Review History**

## Translational perspective

The age-related loss of muscle mass and strength, named sarcopenia, has very important implications for human health, prompting numerous attempts to find effective treatments. Among these, dietary supplementation is particularly appealing being non-invasive and easy to implement. Nitric oxide (NO), a small molecule modulating a plethora of cellular functions, including skeletal muscle perfusion, force production and mitochondrial biogenesis, declines in ageing. Dietary nitrate ($NO_3^-$) supplementation was shown to preserve muscle function in older adults. Such beneficial effects were attributed to a positive modulation of muscle $Ca^{2+}$ release and/or sensitivity, although the underlying mechanisms remain debated. Denervation is considered a major driver of sarcopenia. We show that 2 months of nitrate supplementation in aged mice increased NO availability, blunted ageing-induced NMJ instability and muscle fibre atrophy, enhanced mitochondrial efficiency and lowered mitochondrial ROS production. Such findings suggest potential mechanisms for the beneficial effects of $NO_3^-$ supplementation in humans and support prolonged dietary $NO_3^-$ supplementation as a potential long-term nutritional strategy against sarcopenia. Increasing NO bioavailability in early phases of the ageing process (in middle-aged adults) could help mitigate loss of fast-twitch type II fibres, motor unit remodelling and fibres grouping. Additionally, given the larger effect of NO on fast muscles, nitrates could be particularly effective following periods of inactivity when slow-to-fast fibre type shift occurs. Thus, future studies could explore whether prolonged dietary $NO_3^-$ supplementation may be a relevant approach to mitigate sarcopenia, especially following immobilization/disuse.

