## [Peer Review History · The Journal of Physiology]

Dietary nitrate supplementation mitigates age-related changes at the neuromuscular junction in mice

Maira Rossi, Lucrezia Zuccarelli, Lorenza Brocca, Cristiana Sazzi, Clarissa Gissi, Paola Rossi, Bruno Grassi, Simone Porcelli, Roberto Bottinelli, and Maria Antonietta Pellegrino

DOI: 10.1113/JP287592

Corresponding author(s): Maira Rossi (maira.rossi@unipv.it)

The following individual(s) involved in review of this submission have agreed to reveal their identity: David C Poole (Referee #1); Scott K. Ferguson (Referee #2)

Review Timeline:

Submission Date:	16-Sep-2024
Editorial Decision:	21-Oct-2024
Revision Received:	31-Dec-2024
Accepted:	06-Feb-2025

Senior Editor: Richard Carson

Reviewing Editor: Christoph Centner

Transaction Report:

Dear Dr Rossi,

Re: JP-RP-2024-287592 "Dietary nitrate supplementation mitigates age-related changes at the neuromuscular junction in mice" by Maira Rossi, Lucrezia Zuccarelli, Lorenza Brocca, Cristiana Sazzi, Clarissa Gissi, Paola Rossi, Bruno Grassi, Simone Porcelli, Roberto Bottinelli, and Maria Antonietta Pellegrino

Thank you for submitting your manuscript to The Journal of Physiology. It has been assessed by a Reviewing Editor and by 2 expert referees and we are pleased to tell you that it is potentially acceptable for publication following satisfactory major revision.

REVISION CHECKLIST:

We look forward to receiving your revised submission.

Yours sincerely,

Richard Carson
Senior Editor
The Journal of Physiology

REQUIRED ITEMS

- Author photo and profile. First or joint first authors are asked to provide a short biography (no more than 100 words for one author or 150 words in total for joint first authors) and a portrait photograph. These should be uploaded and clearly labelled together in a Word document with the revised version of the manuscript. See Information for Authors for further details.

- The contact information for the person responsible for 'Research Governance' at your institution needs to be provided. This includes their name and an institutional email address. Please ensure the contact is not an author on this paper and provide an alternate contact if necessary, or confirm in the submission form that the author whose email was provided has sole responsibility for research governance. This is the person who is responsible for regulations, principles and standards of good practice in research carried out at the institution, for instance the ethical treatment of animals, the keeping of proper experimental records or the reporting of results.

- You must start the Methods section with a paragraph headed Ethical approval (https://jp.msubmit.net/cgi-bin/main.plex?form_type=display_requirements#methods).

Research must comply with The Journal's policies regarding animal experiments (<https://physoc.onlinelibrary.wiley.com/hub/animal-experiments>) and adherence to these policies must be stated in the manuscript.

Authors should confirm in their Methods section that their experiments were carried out according to the guidelines laid down by their institution's animal welfare committee, including an ethics approval reference number. The Methods section must contain a statement about access to food, water and housing, details of the anaesthetic regime: anaesthetic used, dose and route of administration, and method of killing the experimental animals.

- Please upload separate high-quality figure files via the submission form.

- Please ensure that any tables are editable and in Word format, and wherever possible, embedded in the article file itself.

- Please ensure that the Article File you upload is a Word file.

- Please include an Abstract Figure file, as well as the Figure Legend text within the main article file. The Abstract Figure is a piece of artwork designed to give readers an immediate understanding of the research and should summarise the main

conclusions. If possible, the image should be easily 'readable' from left to right or top to bottom. It should show the physiological relevance of the manuscript so readers can assess the importance and content of its findings. Abstract Figures should not merely recapitulate other figures in the manuscript. Please try to keep the diagram as simple as possible and without superfluous information that may distract from the main conclusion(s). Abstract Figures must be provided by authors no later than the revised manuscript stage and should be uploaded as a separate file during online submission labelled as File Type 'Abstract Figure'. Please also ensure that you include the figure legend in the main article file. All Abstract Figures should be created using BioRender. Authors should use The Journal's premium BioRender account to export high-resolution images. Details on how to use and access the premium account are included as part of this email.

EDITOR COMMENTS

Reviewing Editor:

The authors present a well designed and well conducted experiment. The data are of potential high relevance for the field. Both reviewers raise important comments and further suggest to implement potential discussions about effects on functional outcomes (except the morphological changes). Additionally, an English proof-read is strongly encouraged to ease the readability of the manuscript.

Please also see 'Required Items' above.

REFEREE COMMENTS

Referee #1:

GENERAL COMMENTS

The demonstration that chronic nitrate supplementation can reverse/prevent crucial sarcopenia events in aged skeletal muscle fills a significant knowledge gap and opens a potentially very exciting avenue for future research. Generally, the experiments were well designed and yielded a plethora of novel data that provide important mechanistic clues regarding the connection between elevated muscle nitrate/nitrite concentrations and mitochondrial, redox, anabolic/catabolic and associated processes. Criticisms relate primarily to a lack of functional data (muscle performance/ exercise performance; probably a future imperative), lack of a supplemented young group (minor), need for streamlining the presentation and writing, some missing technical details, lack of key data in the abstract that predicate the conclusions, lack of consideration for increased muscle oxygenation secondary to nitrate supplementation as a driver of the changes reported. A more refined discussion of Type II errors resulting from the low n per group would also be appropriate especially for Figure 10 mitochondrial control data. That said, most of these shortcomings can be corrected with an excellent revision and there is considerable enthusiasm for this paper and the substantial advance(s) it makes to the field.

SPECIFIC COMMENTS

PAGE/PARA/LINE(S)

1 List specific muscles in key words

2/3/5 Please specify "age-related alterations"

3 Graphical abstract is good. Perhaps also give water to Young mice

4-6 Please refine the writing and reduce by 0.5-0.75 pages.

4/1/3,4 Suggest "were extensively studied"

5/2/10,11 Rewrite for clarity and correctness.

6/2/1 "supplementation would have..."

6/2/4 "the main"

6/2/5 Replace "several" with "related"

7/3 Any calculation of amount actually drunk? Or blood nitrite levels?

7/4/last Please give fiber type composition of muscles used and their citrate synthase activity here rather than in Discussion.

9/2/4 Please state the PO₂ as well. Bailey et al. (2019) demonstrated that some of the acute effects of nitrate/nitrite supplementation are not evident at supra-physiologic PO₂'s.

Bailey SJ, et al. Incubation with sodium nitrite attenuates fatigue development in intact single mouse fibres at physiological PO₂. *J Physiol*. 2019 Nov;597(22):5429-5443. doi: 10.1113/JP278494. Epub 2019 Oct 30. PMID: 31541562; PMCID: PMC6938685.

21/4 In Methods please state how it was ensured that fibers were sectioned in rigorous transverse section? The method used by Mathieu-Costello and colleagues was to alter the sectioning angle to minimize sarcomere counts. Non-orthogonal sections will artifactually elevate measured XSA

21/5/2 Rewrite for clarity

22/2/5,6 But CS activity and mitochondrial volume can be far higher in some fast muscles than the soleus

24/6/5 Somewhere please discuss how OXPHOS coupling efficiency is higher in ON than O when NO inhibits cytochrome oxidase

27/3 How do these changes gel with what is known to happen in aged humans?

28/4 Is eNOS not important?

29/3 "reach" might be better as stipulating "increased nitrate/nitrite"

29/3/6 Delete "be"

31/1/1 "comparable" better as "similar"

Somewhere in the Introduction/discussion some consideration of how nitrate supplementation elevates muscle blood flow and PO₂, during contractions, especially in fast twitch muscles, would be important when considering mechanisms (see Ferguson et al. 2013ab)

Ferguson SK, et al. Impact of dietary nitrate supplementation via beetroot juice on exercising muscle vascular control in rats. *J Physiol*. 2013a Jan 15;591(2):547-57. doi: 10.1113/jphysiol.2012.243121. Epub 2012 Oct 15. PMID: 23070702; PMCID: PMC3577528.

Ferguson SK, et al. Effects of nitrate supplementation via beetroot juice on contracting rat skeletal muscle microvascular oxygen pressure dynamics. *Respir Physiol Neurobiol*. 2013b Jul 1;187(3):250-5. doi: 10.1016/j.resp.2013.04.001. Epub 2013 Apr 11. PMID: 23584049; PMCID: PMC3753182.

Referee #2:

This investigation provides evidence to support the use of dietary nitrate supplementation as a means to preserve muscle function through aging. Using mice, the authors have performed a series of eloquent experiments to demonstrate the effects of 2-month dietary nitrate supplementation on muscle morphology, mitochondrial function, and NMJ deterioration. While the work contributed in this investigation certainly adds to our understanding of nitrate's impact on muscle through the aging process, several considerations should be addressed to improve the clarity and provide context for these findings. Please see my detailed comments/suggestions for improvement below.

1. A major flaw of this manuscript is the copy-editing/language issues present throughout the document. These are mostly minor, however they accumulate over time and distract the reader from the main content of the investigation (e.g., the science). For example, the word assumption is incorrectly used instead of consumption, nitric oxide has been spelled out after it has originally been defined as NO, and the word gently has been used instead of generously. I suggest the authors proofread the document or use a 3rd party service to improve the overall readability of the document.

2. Overall, the experiments conducted in this investigation appear very well planned and executed and do an excellent job of testing the hypothesis. However, given that nitrate supplementation has a stronger impact on type 2 vs. type 1 muscle fibers, and the greater distribution of type 2 muscle fibers in mice vs. humans, what is the likelihood that these findings translate to

humans? Discussion of this limitation is warranted, and adding a clinical relevance section might help guide the reader through these issues, further increasing the translatability of your work.

3. A significant limitation of this investigation is the lack of any functional endpoint to assess whether nitrate supplementation improved muscle function in the aged mice. Measurements of exercise capacity or muscle contractile properties would further support the cellular and molecular findings presented herein. Yes, CSA is greater in mice provided nitrate, but whether this translates to improved muscle function is unclear. Acknowledgment of this limitation and a call to action for future investigations would help put the findings of this investigation into a more explicit context for the reader.

4. The ecological relevance of the dosage of nitrate provided to the mice is unclear. How does the 1.5 mmol of nitrate translate to a dose needed in humans? Does the 1.5 mmol result in plasma levels of nitrate/nitrite increases similar to those seen in humans following a similar dose? Also, please provide a more precise description of the dose and supplementation duration within the abstract, as this is point is unclear and necessary for the reader. Finally, is it 1.5 mmol of nitrate per day or...?

5. Finally, be careful with word choice when describing results collected from multiple groups. More specifically, saying that nitrate increased or decreased anything is incorrect since these are not within animal comparisons. Instead, nitrate mice had significantly greater or lower (insert result here) when compared to old or young mice. This is a minor issue and one I've struggled with in the past, but it certainly should be rectified moving forward.

END OF COMMENTS

EDITOR COMMENTS

Reviewing Editor:

The authors present a well-designed and well conducted experiment. The data are of potential high relevance for the field. Both reviewers raise important comments and further suggest to implement potential discussions about effects on functional outcomes (except the morphological changes). Additionally, an English proof-read is strongly encouraged to ease the readability of the manuscript.

AU: We wish to thank the editor and the reviewers for the very careful revision which enabled us to greatly improve our work. We have tried to address all reviewer's comments. For clarity, we report the number of the page in the revised manuscript at which required revisions were made in response to specific comments of the reviewers. The revisions made to improve the readability of the manuscript are detailed in our response.

REFEREE COMMENTS

Referee #1:

GENERAL COMMENTS

*REF: The demonstration that chronic **nitrate supplementation** can reverse/prevent crucial sarcopenia events in aged skeletal muscle fills a significant knowledge gap and opens a potentially very exciting avenue for future research. Generally, the experiments were well designed and yielded a plethora of novel data that provide important mechanistic clues regarding the connection between elevated muscle nitrate/nitrite concentrations and mitochondrial, redox, anabolic/catabolic and associated processes.*

*Criticisms relate primarily to a lack of **functional data** (muscle performance/ exercise performance; probably a future imperative), lack of a **supplemented young group** (minor), need for **streamlining the presentation and writing**, some **missing technical details**, **lack of key data in the abstract** that predicate the conclusions.*

AU: We would like to thank the reviewer for the constructive comments regarding the structure and experimental design of our manuscript. The reviewer comments and criticisms surely improved the readability and scientific relevance of our work.

We acknowledge that the lack of **functional data** represents a limitation of this study. Addressing this issue will be a future imperative to support the translational impact of the present work. This issue is now mentioned in the "limitations of the study" section of the revised version of the manuscript.

Regarding the absence of a **supplemented young group**, we opted to focus on old mice only, as our primary goal was to assess the potential impact of nitrate supplementation in elderly subjects.

Finally, we have thoroughly revised and proofread the manuscript to improve clarity and **presentation**, and have added additional **technical details**. Changes were made throughout the manuscript, including the abstract, where key data have been incorporated.

*REF: lack of consideration for **increased muscle oxygenation** secondary to nitrate supplementation as a driver of the changes reported. A more refined discussion of **Type II errors** resulting from the low n per group would also be appropriate especially for Figure 10 mitochondrial control data. That said, most of these shortcomings can be corrected with an excellent revision and there is considerable enthusiasm for this paper and the substantial advance(s) it makes to the field.*

AU: We thank the reviewer for the relevant comment regarding potential **Type II errors** due to the small sample size. We acknowledge that the limited number of animals could reduce statistical power and increase the risk of failing to detect true effects, particularly in the mitochondrial control data presented in Figure 10. To mitigate such risk, each animal was tested twice on the same day, and the duplicates were averaged. This information is now included in the Methods section (“*Ex vivo* mitochondrial High-Resolution Respirometry” (pag. 8) and the risk of Type II errors is discussed in “Limitations of the study” section of the revised manuscript (pag. 31). Additionally, we emphasize the need for further studies with larger sample sizes to validate our findings and strengthen the conclusions drawn from the present data.

AU: Thank you for pointing out the potential relevance of **increased muscle oxygenation** secondary to nitrate supplementation as a driver of the observed changes. The text was revised based on the considerations reported below (see “Mechanisms mediating the effects of NO₃⁻ supplementation” pag. 28).

Beetroot juice supplementation is known to improve **vascular control and skeletal muscle O₂ delivery** in both animals and humans (Ferguson *et al.*, 2013a, b; Bailey *et al.*, 2015).

Since cellular energy and oxygen levels are both regulators of the **synthetic pathway** (Saxton & Sabatini, 2017), it could be that nitrate supplementation optimises the cellular environment, thereby providing the energy required for the activation of anabolic pathways, thus enhancing protein synthesis.

Following combined nitrate administration and exercise in old rats, muscle mass, fibre diameter and cell proliferation were enhanced, while exercise alone did not affect muscle mass (Leiter *et al.*, 2012). The capacity of satellite cells to respond to stretch and exercise stimuli is known to be impaired in age-related muscle atrophy. Although the mechanisms underlying the effects of NO are still unsettled, its ability to influence satellite cells activation suggests that the restoration of **satellite cells** regenerative capacity may contribute to muscle mass maintenance following NO administration (Leiter *et al.*, 2012). Interestingly, nitrate administration alone was shown to enhance VEGF expression in old mice, suggesting a contribution of improved vascularisation to the phenomenon (Leiter *et al.*, 2012).

The improvement in vascular control and skeletal muscle O₂ delivery during exercise is **predominant in fast-twitch muscles**, where the reduction of NO₂⁻ to NO is potentiated by a low O₂ environment (Ferguson *et al.*, 2013b). As muscle phenotype is generally slower in humans than in mice, the nitrate-induced enhancement of O₂ delivery is expected to be less marked in humans than in mice. However, strong evidence suggests that nitrate supplementation enhances submaximal oxygen uptake, muscle oxygenation and exercise tolerance in humans, indicating that the slower muscle fibres phenotype does not significantly hinder the efficacy of nitrate on muscle itself.

1 List specific muscles in key words

AU: The request has been addressed.

2/3/5 Please specify "age-related alterations"

AU: The request has been addressed.

3 Graphical abstract is good. Perhaps also give water to Young mice

AU: We thank the reviewer. The request has been addressed and the graphical abstract with the abstract figure legend can now be found attached to the Article File.

4-6 Please refine the writing and reduce by 0.5-0.75 pages.

AU: We refined the "Introduction" and we did our best to shorten it.

4/1/3,4 Suggest "were extensively studied"

AU: The request has been addressed.

5/2/10,11 Rewrite for clarity and correctness.

AU: The request has been addressed.

6/2/1 "supplementation would have..."

AU: The request has been addressed and corrected within the refinement of the paragraph.

6/2/4 "the main"

AU: The paragraph has been refined to improve general clarity.

6/2/5 Replace "several" with "related"

AU: The request has been addressed.

7/3 Any calculation of amount actually drunk? Or blood nitrite levels?

AU: We thank the reviewer for asking to clarify an important point. We have revised the Methods section "NO₃⁻ supplementation" (pag. 6) to provide more detailed information.

The dosage mentioned in the Methods section (1.5 mM NaNO₃) refers to the daily intake of NaNO₃ per mouse, calculated based on mice body weight and the average daily NaNO₃ consumption of old mice. The average daily consumption, previously determined in a sample group of 22-months old mice, was approximately 6 ml/die. Throughout the experimental period (2 months), both body weight and water consumption were monitored weekly.

The selected dosage is based on the study by Hernández et al (Hernandez *et al.*, 2012), in which mice were daily given 1 mM NaNO₃ in their drinking water. This dose was extrapolated from human exercise studies and is comparable to what can be easily achieved through a normal human diet (i.e. daily ingestion of three to four beetroots or 200–300 g of spinach) (Hernandez *et al.*, 2012). The dosage was subsequently adjusted to 1.5 mM (~8 μmol nitrate/day) based on recent evidence suggesting age-related modulation of oral nitrate-reductase activity, which can reduce nitrate/nitrite blood concentrations in both humans and mice (Ahmed *et al.*, 2021).

Unfortunately, we were unable to collect blood samples and measure nitrate/nitrite plasma levels in this study. However, we will surely do the extra effort to include these measurements in future experiments.

7/4/last Please give fiber type composition of muscles used and their citrate synthase activity (here rather than in Discussion)

AU: We thank the reviewer for the suggestion. In the revised version of the manuscript, the "Sample collection" section of the Methods now includes details on MHC isoform distribution and CS activity in different muscle fibres as well as whole muscles (pag. 7).

9/2/4 Please state the PO₂ as well. Bailey et al. (2019) demonstrated that some of the acute effects of nitrate/nitrite supplementation are not evident at supra-physiologic PO₂'s.

AU: We appreciate the reviewer's observation regarding the oxygen conditions during the experiment. We have clarified this point in the "Ex vivo mitochondrial High-Resolution Respirometry" section of the Methods (pagg. 8-9).

The PO₂ range that is maintained throughout the experiment (between 250 and 400 nmol*mL⁻¹, which refers to an average O₂ partial pressure of 250 mmHg) is essential for optimal mitochondrial respiration measurement in permeabilized muscle fibres. We are aware that it is supra-physiologic, and we keep such issue in mind anytime we reason on our data. However, supra-physiologic condition is necessary to avoid oxygen limitation, which can occur due to spatial constraints in fibre preparations, as highlighted by Scandurra and Gnaiger (Scandurra & Gnaiger, 2010). Below air-level oxygen pressures, there is a risk of hypoxic core development. As such, we maintain supra-physiologic conditions to ensure stable and reliable oxygen flux measurements throughout the experiment. This approach is consistent with recommendations in the literature and in the instrument guidelines (OROBOROS Instruments).

Regarding the impact of supra-physiologic PO₂ on the acute effects of nitrate/nitrite supplementation shown by Bailey et al. (Bailey *et al.*, 2019), we are willing to elaborate on such issue if required. However, we decided not to include such point in the current revision as we did not study nitrate/nitrite acute effects on mitochondrial function, but the medium-long term effect of two months supplementation.

21/4 In Methods please state how it was ensured that fibers were sectioned in rigorous transverse section? The method used by Mathieu-Costello and colleagues was to alter the sectioning angle to minimize sarcomere counts. Non-orthogonal sections will artifactually elevate measured XSA

AU: We appreciate the reviewer's comment that enable us to clarify a relevant technical issue. The request has been addressed in the "Immunohistochemistry" paragraph of the Methods (pag. 11). To avoid non-orthogonal sections, which may introduce artifacts in CSA measurements and misrepresent fibre morphology, we focused on the mid-belly region of the Gastrocnemius, where muscle fibres are more uniformly oriented. Furthermore, to verify the accuracy of the transverse orientation sections, we calculated the form factor ($4\pi A/P^2$) (A, cross-sectional area and P, perimeter) for individual fibres using ImageJ software, according to (Charifi *et al.*, 2004). Only fibres with a value greater than 0.7 were considered for the CSA calculation.

21/5/2 Rewrite for clarity

AU: The request has been addressed (now pag. 18).

22/2/5,6 But CS activity and mitochondrial volume can be far higher in some fast muscles than the soleus

AU: We thank the reviewer for the observation. We agree that the relationship between fibre type composition and metabolic properties (e.g. CS activity and mitochondrial volume) does not necessarily match. This point is now discussed in the Methods (pag. 7). However, a mitochondrial content in Gas that is potentially higher than expected, based on fibre type distribution, would not hinder the information we would like to convey in the paragraph, which is that in the old mice used, the expected fast-to-slow shift in muscle fibre type distribution occurs.

24/6/5 Somewhere please discuss how OXPHOS coupling efficiency is higher in ON than O when NO inhibits cytochrome oxidase

AU: We thank the reviewer for the comment regarding the higher OXPHOS coupling efficiency observed in ON compared to O mice. The manuscript has been revised accordingly (pag. 29).

Compared to untreated older mice, the improvement in coupling efficiency ((OXPHOS-LEAK)/OXPHOS) observed in older mice supplemented with nitrates is mainly due to a reduction in LEAK respiration, although not statistically significant, and coupled with unchanged/not significantly changed OXPHOS. LEAK respiration is maintained to compensate for proton leak and corresponds to the maximum protonmotive force. The decrease in this state of respiration suggests a diminished protonmotive force. This aligns with the reduced concentration of mitochondrial ROS levels observed in treated mice, as mitochondrial H₂O₂ is sensitive to changes in protonmotive force. The inhibitory effect of nitrate supplementation on cytochrome oxidase, responsible for transferring electrons from complex III to complex IV, is not observed in the LEAK state since only complexes I and II are stimulated. Similarly, this effect is not present in OXPHOS.

Interestingly, lower LEAK respiration and higher mitochondrial efficiency following nitrate supplementation was also observed in mitochondria from human Vastus Lateralis muscle (Larsen *et al.*, 2011).

We have implemented the Discussion section (pag. 29) and added the formula for the calculation of coupling efficiency in the legend of Figure 10 of the section “Figure legends”.

27/3 How do these changes gel with what is known to happen in aged humans?

AU: We thank the reviewer for raising such relevant issue. In response, we have revised the manuscript to include this discussion point in the “Mitochondrial markers and function” paragraph (pag. 25).

Ageing studies performed on human skeletal muscle indicate a reduction in mitochondrial density, which suggests a general decline in mitochondrial biogenesis with age (Conley *et al.*, 2000; Crane *et al.*, 2010; Peterson *et al.*, 2012). However, the available data on markers of mitochondrial biogenesis and dynamics in humans are limited and often inconsistent.

Some studies have reported a reduction of citrate synthase (CS) activity and a decline in the mRNA levels of PGC-1 α , along with a decrease in the protein levels of Nrf1 and Tfam, in older individuals (Crane *et al.*, 2010; Ghosh *et al.*, 2011). In contrast, other studies indicate that CS activity and OXPHOS complexes are preserved (Johannsen *et al.*, 2012; Wyckelsma *et al.*, 2017), as well as preserved biomarkers of mitochondrial biogenesis. In particular, the levels of SIRT1 and AMPK phosphorylation were found to be unaltered in elderly individuals. Notably, PGC-1 α protein levels were observed to be significantly elevated in elderly individuals (Johannsen *et al.*, 2012).

Regarding mitochondrial dynamics, reduced levels of transcripts for Mfn2 and DRP1 have been observed in muscle biopsies from elderly subjects (Crane *et al.*, 2010). However, studies investigating the protein expression of Mfn2 in older individuals have demonstrated that mitochondrial dynamics proteins, including Mfn2, are upregulated with age (Wyckelsma *et al.*, 2017). This upregulation has been proposed as a protective mechanism against mitochondrial dysfunction. Interestingly, the scenario described in Wyckelsma *et al.* is analogous to what we observed in our animal model. This comment has been incorporated in the Discussion of the revised manuscript.

28/4 Is eNOS not important?

AU: We thank the reviewer for pointing out that we did not discuss the reason why we focused on nNOS. We have revised the manuscript to discuss this point (pag. 27).

At the skeletal muscle level, neuronal NOS is considered the main isoform responsible for NO production (Tengan *et al.*, 2012). We decided to focus on the study of this NOS isoform for its specific

localization within muscle tissue and its structural role. Indeed, nNOS has been shown to interact directly with dystrophin at the level of the sarcolemma.

In contrast, endothelial NOS is constitutively present in skeletal muscle tissue, but mainly expressed in cells which share an endothelial/epithelial identity, making this isoform not as functionally central to NO production in skeletal muscle as nNOS. Therefore, the decision to focus on nNOS is driven by its functional relevance and its structural role in skeletal muscle tissue.

29/3 "reach" might be better as stipulating "increased nitrate/nitrite"

AU: The request has been addressed.

29/3/6 Delete "be"

AU: The request has been addressed.

31/1/1 "comparable" better as "similar"

AU: The request has been addressed.

REF: Somewhere in the Introduction/discussion some consideration of how nitrate supplementation elevates muscle blood flow and PO₂, during contractions, especially in fast twitch muscles, would be important when considering mechanisms (see Ferguson et al. 2013ab).

AU: Thank you for pointing out this important issue. We now discuss this point in “Mechanisms mediating the effects of NO₃⁻ supplementation” (pag. 30).

Referee #2:

REF: This investigation provides evidence to support the use of dietary nitrate supplementation as a means to preserve muscle function through aging. Using mice, the authors have performed a series of eloquent experiments to demonstrate the effects of 2-month dietary nitrate supplementation on muscle morphology, mitochondrial function, and NMJ deterioration. While the work contributed in this investigation certainly adds to our understanding of nitrate's impact on muscle through the aging process, several considerations should be addressed to improve the clarity and provide context for these findings. Please see my detailed comments/suggestions for improvement below.

AU: We would like to thank the reviewer for the constructive comments regarding the structure and experimental design of our manuscript. We appreciate the insights provided during the review process, and we believe that the significant implementation of the manuscript driven by your observation will benefit the whole readability and scientific relevance of our work.

1. A major flaw of this manuscript is the copy-editing/language issues present throughout the document. These are mostly minor, however they accumulate over time and distract the reader from the main content of the investigation (e.g., the science). For example, the word assumption is incorrectly used instead of consumption, nitric oxide has been spelled out after it has originally been defined as NO, and the word gently has been used instead of generously. I suggest the authors proofread the document or use a 3rd party service to improve the overall readability of the document.

AU: We would like to thank the reviewer for pointing out we needed to improve the readability of our manuscript. We carefully revised and proofread the text. We appreciate the suggestions given and believe that revisions surely contributed to the quality and readability of our manuscript.

2. Overall, the experiments conducted in this investigation appear very well planned and executed and do an excellent job of testing the hypothesis. However, given that nitrate supplementation has a stronger impact on type 2 vs. type 1 muscle fibers, and the greater distribution of type 2 muscle fibers in mice vs. humans, what is the likelihood that these findings translate to humans? Discussion of this limitation is warranted, and adding a clinical relevance section might help guide the reader through these issues, further increasing the translatability of your work.

AU: We thank the reviewer for pointing out a relevant issue we did not discuss. We have revised the manuscript to address such point (pag. 30).

The improvement in vascular control and skeletal muscle O₂ delivery during exercise is particularly pronounced in muscles with a fast-twitch phenotype, where the reduction of NO₂⁻ to NO is potentiated by a low O₂ environment (Ferguson *et al.*, 2013b). Since human muscles typically exhibit a slower phenotype compared to mice, the nitrate-induced enhancement of O₂ delivery is expected to be less marked in humans than in mice. However, there is strong evidence that nitrate supplementation enhances submaximal oxygen uptake, muscle oxygenation and exercise tolerance in humans. These findings suggest that the slower muscle fibre phenotype in humans does not significantly hinder nitrate efficacy on muscle itself.

In this regards, we have added an “Author’s Translational Perspective” section to clarify the potential translational impact of our work (pag. 45).

3. A significant limitation of this investigation is the lack of any functional endpoint to assess whether nitrate supplementation improved muscle function in the aged mice. Measurements of exercise capacity or muscle contractile properties would further support the cellular and molecular findings presented herein. Yes, CSA is greater in mice provided nitrate, but whether this translates to improved muscle function is unclear. Acknowledgment of this limitation and a call to action for future investigations would help put the findings of this investigation into a more explicit context for the reader.

AU: We thank the reviewer for the comment. We agree that the lack of functional data could limit the enthusiasm about the translational impact of the work. We now comment on this issue in the “Limitations of the study” (pag. 31).

4. The ecological relevance of the dosage of nitrate provided to the mice is unclear. How does the 1.5 mmol of nitrate translate to a dose needed in humans? Does the 1.5 mmol result in plasma levels of nitrate/nitrite increases similar to those seen in humans following a similar dose? Also, please provide a more precise description of the dose and supplementation duration within the abstract, as this is point is unclear and necessary for the reader. Finally, is it 1.5 mmol of nitrate per day or...?

AU: We thank the reviewer for the thoughtful observations. In our study, the nitrate supplementation provided to the mice was 1.5 mM per day, as detailed in comment 7/3. We revised the manuscript and discussed the dose in the “NO₃⁻ supplementation” paragraph of the Methods (pag. 6).

Regarding the plasma concentration of nitrates/nitrites, unfortunately, we were unable to collect blood for measuring plasma nitrate/nitrite concentrations. However, we expect that, at the same dosage, similar plasma nitrate levels would be achieved in humans and mice, while nitrite levels would be higher in humans. This expectation is based on the findings of Montenegro *et al.* (Montenegro *et al.*, 2016), who compared plasma nitrate and nitrite levels in humans and mice following the administration of the same nitrate dose. These considerations are also reported in Methods of the revised manuscript (pag. 6).

As stated in comment 7/3, we will surely do the extra effort to include these measurements in future experiments.

5. Finally, be careful with word choice when describing results collected from multiple groups. More specifically, saying that nitrate increased or decreased anything is incorrect since these are not within animal comparisons. Instead, nitrate mice had significantly greater or lower (insert result here) when compared to old or young mice. This is a minor issue and one I've struggled with in the past, but it certainly should be rectified moving forward.

AU: We appreciate your insights regarding the clarity of our language in describing results from multiple groups. We were aware of such issue starting from but failed to consider it thoroughly. We have revised the text as suggested.

REFERENCES

- Ahmed KA, Kim K, Ricart K, Van Der Pol W, Qi X, Bamman MM, Behrens C, Fisher G, Boulton ME, Morrow C, O'Neal PV & Patel RP. (2021). Potential role for age as a modulator of oral nitrate reductase activity. *Nitric Oxide* **108**, 1-7.
- Bailey SJ, Gandra PG, Jones AM, Hogan MC & Nogueira L. (2019). Incubation with sodium nitrite attenuates fatigue development in intact single mouse fibres at physiological PO₂. *J Physiol* **597**, 5429-5443.
- Bailey SJ, Varnham RL, DiMenna FJ, Breese BC, Wylie LJ & Jones AM. (2015). Inorganic nitrate supplementation improves muscle oxygenation, O₂ uptake kinetics, and exercise tolerance at high but not low pedal rates. *J Appl Physiol (1985)* **118**, 1396-1405.
- Charifi N, Kadi F, Feasson L, Costes F, Geysant A & Denis C. (2004). Enhancement of microvessel tortuosity in the vastus lateralis muscle of old men in response to endurance training. *J Physiol* **554**, 559-569.
- Conley KE, Jubrias SA & Esselman PC. (2000). Oxidative capacity and ageing in human muscle. *J Physiol* **526 Pt 1**, 203-210.
- Crane JD, Devries MC, Safdar A, Hamadeh MJ & Tarnopolsky MA. (2010). The effect of aging on human skeletal muscle mitochondrial and intramyocellular lipid ultrastructure. *J Gerontol A Biol Sci Med Sci* **65**, 119-128.
- Ferguson SK, Hirai DM, Copp SW, Holdsworth CT, Allen JD, Jones AM, Musch TI & Poole DC. (2013a). Effects of nitrate supplementation via beetroot juice on contracting rat skeletal muscle microvascular oxygen pressure dynamics. *Respir Physiol Neurobiol* **187**, 250-255.
- Ferguson SK, Hirai DM, Copp SW, Holdsworth CT, Allen JD, Jones AM, Musch TI & Poole DC. (2013b). Impact of dietary nitrate supplementation via beetroot juice on exercising muscle vascular control in rats. *J Physiol* **591**, 547-557.
- Ghosh S, Lertwattanak R, Lefort N, Molina-Carrion M, Joya-Galeana J, Bowen BP, Garduno-Garcia Jde J, Abdul-Ghani M, Richardson A, DeFronzo RA, Mandarino L, Van Remmen H & Musi N. (2011). Reduction in reactive oxygen species production by mitochondria from elderly subjects with normal and impaired glucose tolerance. *Diabetes* **60**, 2051-2060.
- Hernandez A, Schiffer TA, Ivarsson N, Cheng AJ, Bruton JD, Lundberg JO, Weitzberg E & Westerblad H. (2012). Dietary nitrate increases tetanic [Ca²⁺]_i and contractile force in mouse fast-twitch muscle. *J Physiol* **590**, 3575-3583.
- Johannsen DL, Conley KE, Bajpeyi S, Punyanitya M, Gallagher D, Zhang Z, Covington J, Smith SR & Ravussin E. (2012). Ectopic lipid accumulation and reduced glucose tolerance in elderly adults are accompanied by altered skeletal muscle mitochondrial activity. *J Clin Endocrinol Metab* **97**, 242-250.

- Larsen FJ, Schiffer TA, Borniquel S, Sahlin K, Ekblom B, Lundberg JO & Weitzberg E. (2011). Dietary inorganic nitrate improves mitochondrial efficiency in humans. *Cell Metab* **13**, 149-159.
- Leiter JR, Upadhaya R & Anderson JE. (2012). Nitric oxide and voluntary exercise together promote quadriceps hypertrophy and increase vascular density in female 18-mo-old mice. *Am J Physiol Cell Physiol* **302**, C1306-1315.
- Montenegro MF, Sundqvist ML, Nihlen C, Hezel M, Carlstrom M, Weitzberg E & Lundberg JO. (2016). Profound differences between humans and rodents in the ability to concentrate salivary nitrate: Implications for translational research. *Redox Biol* **10**, 206-210.
- Peterson CM, Johannsen DL & Ravussin E. (2012). Skeletal muscle mitochondria and aging: a review. *J Aging Res* **2012**, 194821.
- Saxton RA & Sabatini DM. (2017). mTOR Signaling in Growth, Metabolism, and Disease. *Cell* **168**, 960-976.
- Scandurra FM & Gnaiger E. (2010). Cell respiration under hypoxia: facts and artefacts in mitochondrial oxygen kinetics. *Adv Exp Med Biol* **662**, 7-25.
- Tengan CH, Rodrigues GS & Godinho RO. (2012). Nitric oxide in skeletal muscle: role on mitochondrial biogenesis and function. *Int J Mol Sci* **13**, 17160-17184.
- Wyckelsma VL, Levinger I, McKenna MJ, Formosa LE, Ryan MT, Petersen AC, Anderson MJ & Murphy RM. (2017). Preservation of skeletal muscle mitochondrial content in older adults: relationship between mitochondria, fibre type and high-intensity exercise training. *J Physiol* **595**, 3345-3359.

Dear Dr Rossi,

Re: JP-RP-2024-287592R1 "Dietary nitrate supplementation mitigates age-related changes at the neuromuscular junction in mice" by Maira Rossi, Lucrezia Zuccarelli, Lorenza Brocca, Cristiana Sazzi, Clarissa Gissi, Paola Rossi, Bruno Grassi, Simone Porcelli, Roberto Bottinelli, and Maria Antonietta Pellegrino

We are pleased to tell you that your paper has been accepted for publication in The Journal of Physiology.

Yours sincerely,

Richard Carson
Senior Editor
The Journal of Physiology

If you would like to receive our 'Research Roundup', a monthly newsletter highlighting the cutting-edge research published in The Physiological Society's family of journals (The Journal of Physiology, Experimental Physiology, Physiological Reports, The Journal of Nutritional Physiology and The Journal of Precision Medicine: Health and Disease), please click this link, fill in your name and email address and select 'Research Roundup':
<https://www.physoc.org/journals-and-media/membernews>

- You can help your research get the attention it deserves! Check out Wiley's free Promotion Guide for best-practice recommendations for promoting your work at: www.wileyauthors.com/eeo/guide. You can learn more about Wiley Editing Services which offers professional video, design, and writing services to create shareable video abstracts, infographics, conference posters, lay summaries, and research news stories for your research at: www.wileyauthors.com/eeo/promotion.

EDITOR COMMENTS

Reviewing Editor:

We congratulate the authors on a well improved manuscript

REFEREE COMMENTS

Referee #1:

The authors have undertaken a substantial revision that has addressed the principal concerns raised. The authors are to be congratulated on improving the clarity and completeness of this manuscript and on the putative high impact of this exciting work. A personal quibble remains the lack of quantitative data in the Abstract to substantiate the conclusions. However, I am happy to leave that to the editor and authors' discretion.

Referee #2:

The authors have done an excellent job addressing my original comments/suggestions. Thank you for your contribution to the field!